# CKAP5 stabilizes CENP-E at kinetochores by regulating microtubule-chromosome attachments

R Bhagya Lakshmi[1], Pinaki Nayak[2,5], Linoy Raz[3,5], Apurba Sarkar [ID][2], Akshay Saroha [ID][4], Pratibha Kumari[4], Vishnu M Nair[1], Delvin P Kombarakkaran[1], S Sajana [ID][1], Sanusha M G[1], Sarit S Agasti[4], Raja Paul[2], Uri Ben-David [ID][3] & Tapas K Manna [ID][1✉]

## Abstract

Stabilization of microtubule plus end-directed kinesin CENP-E at the metaphase kinetochores is important for chromosome alignment, but its mechanism remains unclear. Here, we show that CKAP5, a conserved microtubule plus tip protein, regulates CENP-E at kinetochores in human cells. Depletion of CKAP5 impairs CENP-E localization at kinetochores at the metaphase plate and results in increased kinetochore–microtubule stability and attachment errors. Erroneous attachments are also supported by computational modeling. Analysis of CKAP5 knockout cancer cells of multiple tissue origins shows that CKAP5 is preferentially essential in aneuploid, chromosomally unstable cells, and the sensitivity to CKAP5 depletion is correlated to that of CENP-E depletion. CKAP5 depletion leads to reduction in CENP-E-BubR1 interaction and the interaction is rescued by TOG4-TOG5 domain of CKAP5. The same domain can rescue CKAP5 depletion-induced CENP-E removal from the kinetochores. Interestingly, CKAP5 depletion facilitates recruitment of PP1 to the kinetochores and furthermore, a PP1 target site-specific CENP-E phospho-mimicking mutant gets stabilized at kinetochores in the CKAP5-depleted cells. Together, the results support a model in which CKAP5 controls mitotic chromosome attachment errors by stabilizing CENP-E at kinetochores and by regulating stability of the kinetochore-attached microtubules.

**Keywords** Microtubule; Kinetochore; CKAP5; CENP-E; PP1
**Subject Categories** Cell Adhesion, Polarity & Cytoskeleton; Cell Cycle

## Introduction

Accurate chromosome segregation requires proper attachment of the dynamic spindle microtubule plus ends to the kinetochore (KT), a supramolecular structure assembled on the centromere. Errors in kinetochore–microtubule (KT–MT) attachment compromise the fidelity of chromosome segregation and contribute to the genesis of cancer. During the early mitotic stage, KTs attach laterally to the walls of MTs and are pulled toward the spindle poles through the minus end-directed motor, dynein (Rieder and Alexander, 1990). The pole-proximal chromosomes then move towards the spindle equator and are congressed at the metaphase plate. During this process, the lateral attachments are transformed into end-on attachments with both the sister KTs leading to their bioriented configuration, a requirement for faithful chromosome segregation (Torvi et al, 2022). MT-dependent plus end-directed Kinesin 7 family motor, Centromeric Protein E (CENP-E) plays essential roles both in transporting the pole-proximal chromosomes for congression to the metaphase plate and in maintaining the sustained KT–MT attachment stabilization at the plate thereafter (Gudimchuk et al, 2013; Kim et al, 2010; Schaar et al, 1997; Vitre et al, 2014). Inhibition of CENP-E in the metaphase-congressed state has been shown to delocalize chromosomes from the plate leading to chromosome segregation errors (Gudimchuk et al, 2013).

CENP-E shows a strong localization at the KTs of dispersed uncongressed chromosomes during early mitosis till prometaphase, when KTs are yet to achieve end-on attachment (Magidson et al, 2015; Sacristan et al, 2018). It appears as large crescent-shaped structures at the outer KT primarily due to its association with the outermost fibrous corona layer. As the cells progress to metaphase and the end-on attachment is established, CENP-E along with several other fibrous corona proteins have been shown to be stripped off from the KTs in a dynein-dependent manner (Howell et al, 2001; Sacristan et al, 2018). However, unlike the other fibrous corona proteins, a significant amount of CENP-E is retained at the KTs after the chromosomes are congressed (Brown et al, 1996; Cooke et al, 1997; Gudimchuk et al, 2013). Such persistent presence of CENP-E at the metaphase KTs is essential as it tip tracks both the polymerizing and depolymerizing microtubule ends in order to stabilize KT attachments with the dynamic MT ends at that stage (Gudimchuk et al, 2013). Specifically, it has been shown that pharmacological inhibition of CENP-E at the metaphase-congressed KT stage moves the chromosomes away from the plate toward the pole-proximal region (Gudimchuk et al, 2013). CENP-E

[1]School of Biology, Indian Institute of Science Education and Research, Thiruvananthapuram, Vithura, Thiruvananthapuram, Kerala 695551, India. [2]School of Mathematical and Computational Sciences, Indian Association for the Cultivation of Science, Jadavpur, Kolkata 700032, India. [3]Department of Human Molecular Genetics and Biochemistry, Faculty of Medicine, Tel Aviv University, Tel Aviv, Israel. [4]New Chemistry Unit, Jawaharlal Nehru Centre for Advanced Scientific Research, Bengaluru, Karnataka 560064, India. [5]These authors contributed equally: Pinaki Nayak, Linoy Raz. ✉E-mail: tmanna@iisertvm.ac.in

consists of an MT-binding domain toward the C-terminus and its association with the MTs is essential for its tip tracking activity and its mediated chromosome congression (Gudimchuk et al, 2013). Previous studies also showed that mitotic checkpoint protein BubR1 facilitates CENP-E stabilization at the KTs through interaction and further by inducing its phosphorylation (Chan et al, 1998; Huang et al, 2019; Legal et al, 2020; Mao et al, 2003). CENP-E interacts with protein phosphatase 1 (PP1), a master regulator of spindle assembly checkpoint (SAC) which dephosphorylates several KT proteins and thereby promotes SAC silencing (Emanuele et al, 2008; Kim et al, 2010; Liu et al, 2010; Nijenhuis et al, 2014; Nilsson, 2019; Ruggiero et al, 2020). It has also been shown previously that PP1 dephosphorylates CENP-E presumably at a site specific to Aurora B-mediated phosphorylation and this regulation is critical for the congression of chromosomes to the metaphase plate (Kim et al, 2010). Despite CENP-E's crucial roles in metaphase chromosome congression, the mechanisms of its recruitment to and stabilization at the KT–MT junction remain to be understood.

The XMAP215 family protein Stu2 (Budding yeast) /CKAP5 (Human) is a conserved MT plus end-associated protein that localizes to both KT-attached MT plus ends and KTs and it is essential for error-free KT attachments with MTs (Herman et al, 2020). Depletion of CKAP5 in human cells leads to severe defects in chromosome congression to the metaphase plate (Barr and Bakal, 2015; Barr and Gergely, 2008; Gergely et al, 2003; Herman et al, 2020). Experiments with reconstituted KT particles in vitro showed that under low tension between the sister KTs during erroneous attachments, stu2/CKAP5 destabilizes the attachments; but during correct attachment under high tension, it facilitates MT polymerization to sustain the tension (Miller et al, 2016). Previous studies also showed that CKAP5 depletion induces stabilization of erroneous KT–MT attachments during early mitosis (Herman et al, 2020) and its MT lattice binding is conferred by a conserved Serine-Lysine (S-K) rich region located between the TOG4 and 5 domains (Al-Bassam and Chang, 2011; Miller et al, 2019; Zahm et al, 2021). Mutation of this site stabilizes KT–MT attachments with polar chromosomes similar to that observed in the CKAP5-depleted cells (Herman et al, 2020). Similar to CENP-E, XMAP215 can tip track and associate with both polymerizing and depolymerizing microtubule ends in vitro (Brouhard et al, 2008). Though both CKAP5 and CENP-E are critical for KT–MT attachment error correction and chromosome congression, their functional link remains unclear. Here, we show that CKAP5 facilitates the stabilization of CENP-E at metaphase KTs in human cells. CKAP5 depletion disrupts CENP-E localization at KTs of the partially aligned chromosomes at the metaphase plate and induces kMT (kinetochore-attached microtubules) hyperstabilization and attachment errors, which can result in chromosome mis-segregation and aneuploidy induction. A similar propensity of erroneous attachments was apparent in our computational model. Complementarily, large-scale analyses of hundreds of human cancer cell lines revealed that aneuploid, chromosomally unstable cells are more sensitive to CKAP5 perturbation. Moreover, the mRNA and protein expression levels of CKAP5 and CENP-E are strongly correlated, and increased dependency on CKAP5 is strongly associated with increased dependency on CENP-E, further supporting a functional link between the two proteins and suggesting that aneuploid cells are more dependent on the proper

function of these proteins to prevent excessive chromosomal instability. We further show that recruitment of PP1 is facilitated both during prometaphase and metaphase in CKAP5-depleted cells and further, a PP1 site-specific phospho-mimicking CENP-E mutant gets stabilized at KTs of CKAP5-depleted cells. Together, the results support a CKAP5-CENP-E-dependent KT attachment stabilization and error-free chromosome segregation.

## Results

### CKAP5 is required for the stabilization of CENP-E at kinetochores

We first sought to characterize the chromosomal defects in response to the loss of CKAP5. CKAP5 was depleted by using esiRNA in cultured human cell lines, HeLa Kyoto, and U2OS. CKAP5 level was reduced by ~85% by esiRNA in HeLa Kyoto cells (Fig. EV1C). Depletion of CKAP5 led to severe chromosome congression defects in the majority of the mitotic cells. Consistent with earlier studies (Gergely et al, 2003), ~60% of mitotic cells showed partially aligned chromosomes at the metaphase plate and a few uncongressed chromosomes dispersed near the poles (Fig. EV1A,B). We then looked into the localization of KT proteins specific to different layers of KT in the CKAP5-depleted cells. It revealed that the fibrous corona and outer KT-associated protein, CENP-E was specifically lost from the partially aligned metaphase plate KTs in HeLa Kyoto cells upon depletion of CKAP5 (Fig. 1A,B). KT intensity of CENP-E in HeLa Kyoto cells was reduced by ~85% at the aligned KTs (Fig. 1C). CKAP5 depletion-induced loss of CENP-E was also apparent at the KTs of prometaphase cells (Fig. 1D–F). Prometaphase arrest was induced by treating the cells with Eg5 kinesin inhibitor, DMA (Dimethylenastron) (Tamura et al, 2015). CKAP5 depletion-induced CENP-E mis-localization from KTs was not due to any change of CENP-E expression as the overall cellular level of CENP-E was not altered upon CKAP5 depletion (Fig. EV1D). Similar results were obtained in U2OS cells depleted of CKAP5 by esiRNA and also in inducible Cas-9 expressing CKAP5 knockout HeLa cells (Fig. EV1E–G).

CKAP5 can localize to KTs even in the absence of MTs (Herman et al, 2020). To determine if the abrogation of CENP-E from KTs upon CKAP5 depletion can occur in the absence of microtubules, we checked CENP-E localization in cells by treating with 3.3 μM nocodazole (Sacristan et al, 2018), which completely depolymerizes MTs. KT localization of CENP-E was unaffected in the CKAP5-depleted cells upon depolymerization of the MTs (Fig. 1G), suggesting that CKAP5 depletion-induced CENP-E abrogation from the KTs involves KT-attached microtubules. Dynein plays an important role in removing fibrous corona proteins including CENP-E from MT-attached KTs (Howell et al, 2001; Sacristan et al, 2018). We next checked if CENP-E removal from the partially aligned KTs under CKAP5-depleted condition depends on dynein-mediated stripping by co-depleting dynein adaptor, Spindly along with CKAP5 in HeLa Kyoto cells. CENP-E levels were rescued significantly at the partially aligned KTs in the co-depleted cells, suggesting that CENP-E destabilization in CKAP5-depleted condition is due to CENP-E stripping by dynein (Fig. EV1H–J).

We next sought to identify the region of CKAP5 that could be involved in CENP-E regulation by rescue experiments using

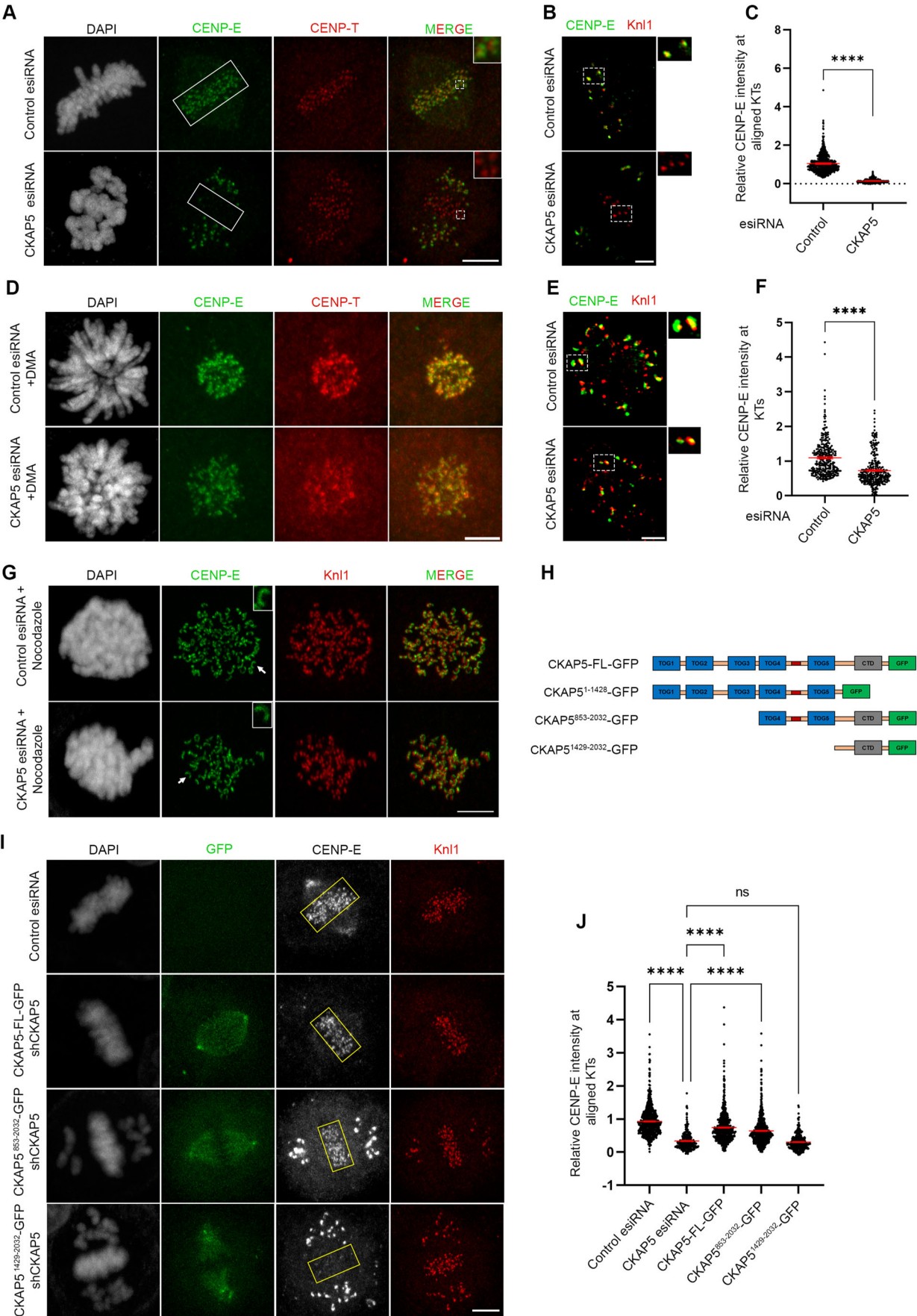

**Figure 1. CKAP5 is essential for CENP-E localization at congressed kinetochores.**

(A) Representative immunofluorescence images showing partially aligned chromosomes of HeLa Kyoto cells after treatment (48 h) with control esiRNA or CKAP5 esiRNA stained for CENP-E (green), CENP-T (red) and DAPI (gray). Scale bar = 5 μm. Regions marked by dotted boxes in the merge images are enlarged in insets. (B) Structured illumination microscopy (SIM) images showing CENP-E (green) localization with respect to Knl1 (red) at the aligned KTs (single plane). Scale bar = 2 μm. Regions in dotted boxes are enlarged in insets. (C) Dot plots showing ratio of mean intensities of CENP-E/CENP-T (from A) at individual KTs that are aligned (for control)/partially aligned (for CKAP5 depleted) at the metaphase plate (shown in boxes) (n ~750 KTs). The intensity values obtained from three independent experiments are plotted. ****P < 0.0001 by Mann–Whitney test. Data represents mean +/− SEM. (D) Representative immunofluorescence images showing prometaphase chromosomes of HeLa Kyoto cells after treatment (48 h) with control esiRNA or CKAP5 esiRNA stained for CENP-E (green), CENP-T (red) and DAPI (gray). Scale bar = 5 μm. (E) Structured illumination microscopy (SIM) images showing CENP-E (green) localization with respect to Knl1 (red) at the prometaphase KTs (single plane). Scale = 2 μm. Regions in dotted boxes are enlarged in insets. (F) Dot plots showing ratio of mean intensities of CENP-E/CENP-T (from D) at individual KTs of prometaphase arrested cells (n ~300 KTs). The intensity values obtained from three independent experiments are plotted. ****P < 0.0001 by Mann–Whitney test. Data represents mean +/− SEM. (G) Nocodazole-treated control or CKAP5-depleted cells showing KT localization of CENP-E (green). Knl1 (red) was used as KT marker. Scale bar = 5 μm. KTs shown with arrows are enlarged in insets. (H) Schematic representation of various truncated CKAP5 proteins fused with C-terminal GFP tag. (I) Immunofluorescence images showing CENP-E (grey) localization at metaphase plate-aligned KTs of un-transfected HeLa Kyoto cells and cells expressing CKAP5-FL, CKAP5$^{853-2032}$ and CKAP5$^{1429-2032}$ with simultaneous endogenous CKAP5-depleted condition. Knl1 (red) was used as KT marker. Scale bar = 5 μm. (J) Dot plot showing mean intensity ratio of CENP-E/Knl1 at plate-aligned (shown in yellow boxes (I)) KTs (n ~700 KTs). ****P < 0.0001 by one-way ANOVA. Data represent mean +/− SEM. Source data are available online for this figure.

various CKAP5 deletion constructs (Fig. 1H). While the full length and CKAP5 consisting of TOG4, TOG5 and the C-terminus (CKAP5$^{853-2032}$) could rescue CENP-E localization at KTs, when co-expressed with CKAP5-shRNA in HeLa Kyoto cells, the C-terminal only domain (CKAP5$^{1429-2032}$) failed to do so (Figs. 1I,J and EV1K). Also, CKAP5$^{853-2032}$ expressing cells showed a reduced number of unaligned polar chromosomes (Fig. EV1L,M). The N-terminal only region of CKAP5 consisting of all the TOG domains (CKAP5$^{1-1428}$) did not show any specific localization to KTs or spindles (Fig. EV1N). Exogenous expression of CKAP5-GFP full length in endogenous CKAP5-depleted prometaphase cells also showed significant rescue of CENP-E localization as compared to only CKAP5-depleted prometaphase cells (Fig. EV2A,B). Taken together, the results indicate that MT-associated CKAP5 plays a critical role in stabilizing CENP-E at the aligned KTs and further, the region containing TOG4 and TOG5 is involved in this function.

## CKAP5 regulates CENP-E–BubR1 interaction

Since CENP-E localization at attached KTs is known to be facilitated by its direct interaction with the Bub complex protein BubR1 (Chan et al, 1998; Huang et al, 2019; Legal et al, 2020; Mao et al, 2003), we next sought to examine CENP-E-BubR1 interaction in the CKAP5-depleted cells. Pulldown of MycGFP-CENP-E (by GFP Trap) from mitotic synchronized HEK 293T cells depleted of endogenous CKAP5 and expressed with MycGFP-CENP-E showed substantially reduced interaction of endogenous BubR1 with MycGFP-CENP-E as compared to the same in control cells (Fig. 2A,B). Depletion of Bub complex proteins (Bub1 and BubR1) has been shown to delocalize CENP-E from the KTs (Johnson et al, 2004; Legal et al, 2020). Therefore, to rule out the possibility that CKAP5 depletion-induced loss of CENP-E from KTs is due to defects in KT localization of BubR1/Bub complex, we checked BubR1 localization in the CKAP5-depleted cells. BubR1 levels at the aligned KTs were either not significantly affected or in some cases slightly increased but not reduced in the CKAP5 knockdown cells (Fig. 2C,D). Similar results were obtained for Bub1 localization as well (Fig. EV2C,D). Furthermore, since CKAP5$^{853-2032}$-GFP could rescue CENP-E localization at the aligned KTs under endogenous CKAP5-depleted condition, we next checked the effect of CKAP5$^{853-2032}$-GFP expression on CENP-E-BubR1 interaction in HEK 293T cells depleted of endogenous CKAP5. CENP-E-BubR1 interaction was substantially rescued upon expression

of CKAP5$^{853-2032}$-GFP in these cells, supporting a direct role of CKAP5 in facilitating CENP-E-BubR1 interaction and thereby stabilizing CENP-E at the KTs (Fig. 2E,F).

## CKAP5 is required for the regulation of spindle microtubule stability

Since MTs are essential for CKAP5 depletion-induced CENP-E abrogation from KTs, we characterized specific MT-associated defects, if any, in the CKAP5-depleted cells. To visualize specifically the KT-attached MTs, the non-KT MTs were depolymerized under cold condition (Wu et al, 2019). Immunofluorescence imaging of CKAP5-depleted cells showed MTs attached to the sister KTs with unequal lengths and reduced inter-KT distance between sister KT pairs of the aligned chromosomes (Fig. 3A–C). It was also revealed that CKAP5 depletion resulted in an increased incidence of the number of cells having at least one merotelically attached KT (Fig. 3D,E), and syntelic and monotelic attachments of the pole-proximal chromosomes (Fig. EV2E,F). Merotelic attachments, if not corrected, generate lagging chromosomes during anaphase (Sacristan et al, 2018). Presence of merotelic attachments was therefore, further confirmed by checking for the incidence of lagging chromosomes during anaphase. For this, CKAP5-depleted cells and control cells were treated with 250 nM Reversine (Mps1 inhibitor) for inducing anaphase (Sacristan et al, 2018) and it was observed that ~73% cells showed lagging chromosomes in anaphase, when CKAP5 was depleted suggesting the prevalence of merotelic attachments (Fig. EV2G,H). Interestingly, a significant number of KT-attached MTs (kMT) of CKAP5-depleted cells showed increased MT density (thicker MTs), suggesting enhanced stabilization of the kMTs (Inset, Fig. 3A,F). However, the increased MT stabilization phenotype was rescued by the expression of CKAP5$^{853-2032}$-GFP which encompasses the basic linker region which is important for its MT lattice binding (Figs. 3G and EV2I). Next, we sought to understand how increased MT stability can induce CENP-E removal from the aligned KTs. For this, MT hyperstabilization and low inter-KT tension phenotypes observed in CKAP5-depleted cells were mimicked in control cells by treating them with 10 μM paclitaxel for 30 min. Interestingly, CENP-E level at aligned KTs under such condition was intact as compared to DMSO-treated cells (Fig. 3H,I), suggesting that MT hyperstabilization alone is not solely responsible for CENP-E removal. Previous

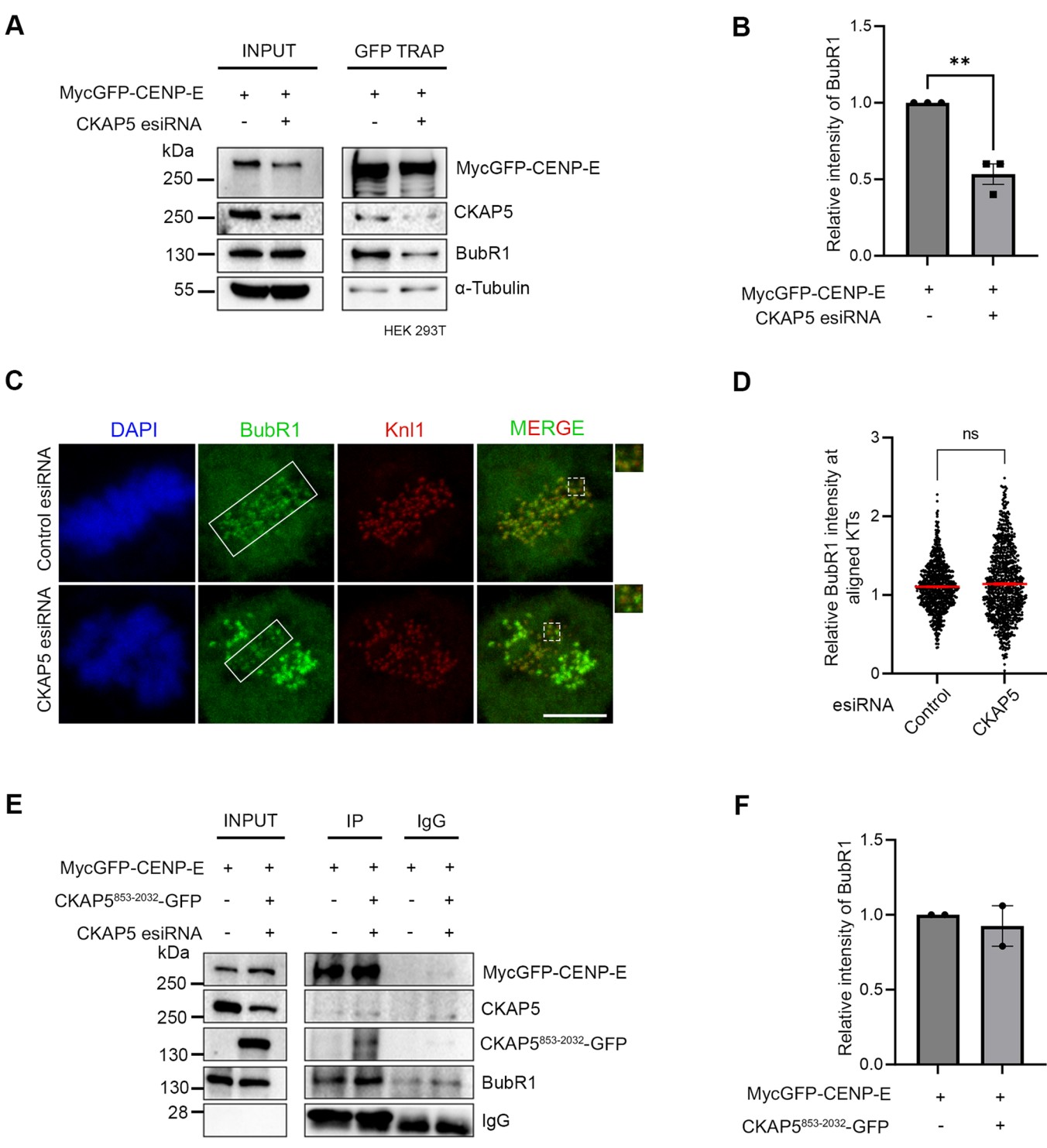

reports suggested that paclitaxel interferes with MT tracking ability of dynein (Forer et al, 2018; Gurden et al, 2018), which could also result in CENP-E stabilization at KTs. The role of MT hyperstability in CENP-E removal was further tested by partially destabilizing MTs in CKAP5-depleted cells using 3.3 µM nocodazole for 10 min (Fig. EV3A). A significant rescue in CENP-E levels at the partially aligned KTs was observed under partial nocodazole condition (Fig. 3J,K). This suggests that MT hyperstabilization in

the absence of CKAP5 has indeed a significant role in CENP-E destabilization. Recent evidence suggests that point mutations (K1142, K1143) at the basic linker region of CKAP5 induce increased stability of microtubules attached to kinetochores of polar chromosomes similar to that observed in CKAP5-depleted cells (Herman et al, 2020). In order to test the role of this mutation on CENP-E levels, HeLa Kyoto cells were expressed with the basic linker region mutant of CKAP5, CKAP5$^{853-2032}$-KK/AA-GFP under

◀ **Figure 2.  CKAP5 regulates CENP-E–BubR1 interaction.**

(A) GFP Trap pulldown from HEK 293T cells expressing MycGFP-CENP-E in endogenous CKAP5-depleted background showing reduced interaction of CENP-E with BubR1. (B) Mean intensity of BubR1 bands normalized with MycGFP-CENP-E bands from three independent experiments were plotted. **$P = 0.0022$, by Student's $t$ test. (C) Representative immunofluorescence image showing no significant change in localization of BubR1 at aligned KTs of control vs. CKAP5-depleted HeLa Kyoto cells (region marked with box in (C)). Insets show enlarged view of region marked by dotted boxes in the merge images. Scale bar = 5 μm. (D) Dot plots showing mean intensities of BubR1/Knl1 ($n$ ~600 KTs) quantified from individual kinetochores of multiple cells from three independent experiments. The difference was shown to be nonsignificant by Student's $t$ test. (E) Myc pulldown from CKAP5-depleted HEK 293T cells expressing MycGFP-CENP-E and CKAP5$^{853-2032}$-GFP showing rescue of CENP-E-BubR1 interaction. Same in cells without CKAP5 depletion is shown as control. (F) Mean intensities of BubR1 bands normalized with MycGFP-CENP-E bands (corresponding to (E)) from two independent experiments are plotted. All data represent mean +/− SEM. Data Information: A significant difference (**$P = 0.0022$) in CENP-E-BubR1 interaction is observed between control and CKAP5-depleted cells ($n = 3$) by Student's $t$ test for (B). No difference in CENP-E–BubR1 interaction is observed between control and CKAP5-depleted cells expressing CKAP5$^{853-2032}$-GFP ($n = 2$). Source data are available online for this figure.

endogenous CKAP5-depleted condition and the KT localization of CENP-E was assessed. The levels of CENP-E at the aligned KTs were significantly reduced in the KK/AA mutant-expressed condition as compared to CKAP5$^{853-2032}$-WT-GFP expression (Fig. 3L,M), but the extend of reduction in CENP-E levels at the KTs of cells expressing CKAP5$^{853-2032}$-KK/AA mutant was significantly less as compared to the near complete removal of CENP-E in CKAP5-depleted condition (Fig. 1A–C). Taken together, the results indicate that CKAP5 mediates CENP-E stabilization at KTs by regulating the spindle microtubule stability.

## CKAP5 regulates PP1 at kinetochores

Previous studies showed that correct KT–MT end-on attachment stimulates PP1 activity at KTs leading to attachment stabilization (Emanuele et al, 2008; Kim et al, 2010; Liu et al, 2010; Nijenhuis et al, 2014). Moreover, PP1 also dephosphorylates CENP-E and supposedly induces its removal from KTs (Kim et al, 2010). Therefore, we checked how PP1 activity at the KTs is regulated in CKAP5-depleted cells. Immunofluorescent staining of PP1 in CKAP5-depleted cells showed its increased localization at the partially aligned metaphase KTs and also at prometaphase KTs (Fig. 4A–D). Consistent with enhanced PP1 recruitment, the level of phosphorylation of the outer KT protein, such as Hec1 showed a concomitant decrease at the KTs in prometaphase cells (Fig. EV3B), suggesting that PP1 activity at the KTs is increased in these cells. Interestingly, expression of CKAP5$^{853-2032}$-GFP which could rescue the enhanced MT stabilization phenotype, could also rescue the levels of PP1 at prometaphase KTs in endogenous CKAP5-depleted cells (Fig. EV3D,E). We then checked whether CENP-E delocalization from KTs in CKAP5-depleted cells could be attributed to increased PP1 activity at the KTs. Treatment of CKAP5-depleted cells with okadaic acid (0.25 μM) which inhibits PP1 activity resulted in nearly complete rescue of CENP-E at the KTs (Fig. 4E,F). Under similar conditions, a lower concentration of okadaic acid (0.12 μM) couldn't rescue CENP-E localization, suggesting that PP1 but not PP2A is responsible for the CENP-E delocalization phenotype in CKAP5-depleted cells (Fig. EV3F). Furthermore, CENP-E-BubR1 interaction was rescued significantly in the CKAP5-depleted condition, when treated with okadaic acid in HEK 293T cells (Fig. 4G,H) substantiating the role of increased PP1 activity for the KT removal of CENP-E under CKAP5 depletion. Initial Aurora A/B-mediated phosphorylation followed by PP1-mediated dephosphorylation at the Aurora B-targeting site, Thr 422 of CENP-E facilitates metaphase chromosome alignment (Kim et al, 2010). To test the possibility that KT removal of CENP-E in CKAP5-depleted cells is associated with its PP1-mediated

dephosphorylation at Thr 422, we checked the KT localization of a phospho-mimetic mutant of CENP-E specific to Thr 422 site (MycGFP-CENP-E-T422E). The KT intensity of CENP-E T422E mutant was checked and compared with that of CENP-E wildtype and a phospho-deficient T422A mutant under endogenous CENP-E depletion by siRNA in doxycycline-induced CKAP5 knockout HeLa cells (Fig. EV3G). Unlike GFP-CENP-E-WT, which was nearly completely lost from the KTs (Fig. 4I), GFP-CENP-E-T422E mutant was significantly stabilized at the partially aligned KTs under doxycycline-induced CKAP5 depletion condition (Fig. 4J,K). As reported in a recent study, the phospho-deficient mutant, CENP-E-T422A, could not localize to the partially aligned KTs under endogenous CENP-E-depleted condition (Eibes et al, 2023) irrespective of the presence or absence of CKAP5 (EV3 H). Since PP1 targeting to CENP-E and presumably PP1-mediated CENP-E dephosphorylation are critical for the removal of CENP-E from the KTs (Eibes et al, 2023; Kim et al, 2010), our results support the model that CKAP5 depletion induced MT over-stabilization, and the associated increased KT recruitment of PP1 could control the level of phosphorylated-CENP-E and its stabilization at the KTs.

## In silico study supports the experimentally observed chromosome congression defects and attachment errors under CKAP5-depleted condition

We developed a basic mechanistic model of kMT–KT (kinetochore-attached microtubule–kinetochore) attachment error correction mechanism to comprehend the role of CKAP5 in establishing the correct end-on attachments. We only considered primary components of the mitotic spindle, including the astral MTs, kMTs, sister KTs, and centrosomes (Fig. 5A). The effect of CENP-E and other fibrous corona proteins, appear in the model as spring-like attachments between kMT tip and the KT (Fig. 5B,C) analogous to earlier studies (Powers et al, 2009; Sutradhar et al, 2015; Thomas et al, 2016). Depletion of CKAP5 leads to the loss of CENP-E at KTs in the experiments, and this is replicated in the model by a reduced strength of the kMT–KT spring-like attachments (Thomas et al, 2016). We assume the rate of the detachment of kMT–KT bond is proportional to the force applied on the kMT–KT springs for syntelic and merotelic attachments (Fig. 5D). We consider the force-dependent turnover of kMT–KT attachments based on earlier models of cargo hauling according to which an increased load force on molecular motors mediated by the cargoes enhances their detachment rate with the underlying MT track (Klumpp and Lipowsky, 2005; Kunwar et al, 2011). In addition, the detachment rate of a merotelic attachment is assumed to be proportional to the angle formed by the kMT–KT attachment with the KT–KT axis,

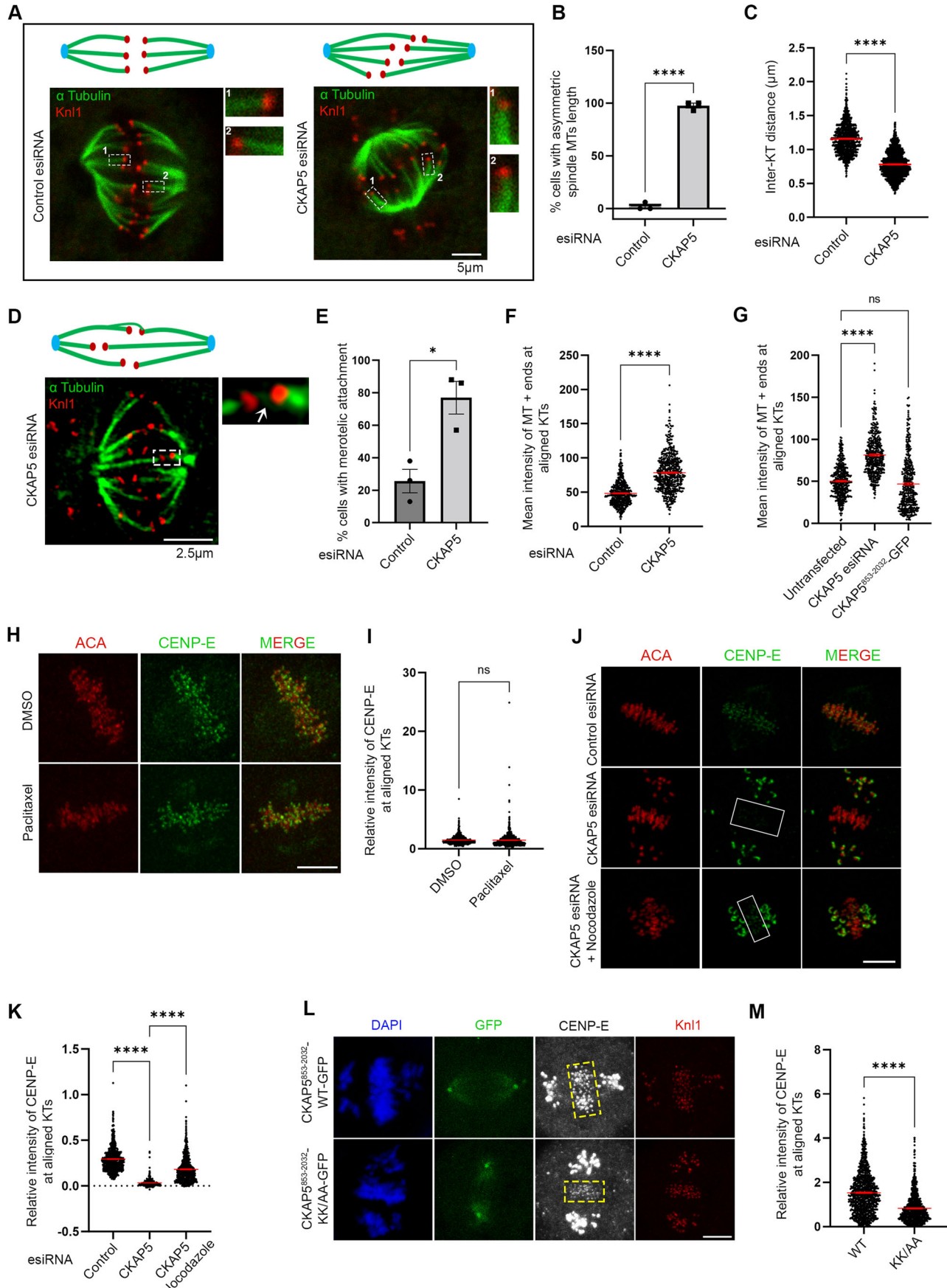

**Figure 3. CKAP5 depletion leads to KT–MT attachment errors.**

(A) Representative immunofluorescence images (single plane) showing control siRNA-treated and CKAP5-depleted HeLa Kyoto cells stained for α-Tubulin (green) and Knl1 (red). Cartoons depicting symmetric and asymmetric spindle MT lengths are shown. Dotted boxes highlighting microtubule thickness are enlarged in the inset. (B) Graph showing the percentage of cells having asymmetric MT lengths quantified from 30 cells. Dots represent the mean value of each experiment ($N = 3$) and bars represent their average. ****$P < 0.0001$ by Student's $t$ test. (C) Average inter- KT distance at aligned KTs of nearly 1000 sister KT pairs together from three independent experiments ****$P < 0.0001$ by Student's $t$ test. (D) Representative structured illumination microscopy (SIM) image of CKAP5-depleted cell showing merotelic attachment error (shown in the dotted box and inset with arrow showing the merotelic attached MT) at aligned KTs. The same is represented by a cartoon on top. α-Tubulin is stained green and Knl1 is stained red. (E) Graph showing the percentage of metaphase cells ($n$ ~30 cells for each) with at least one merotelic attachment in control vs. CKAP5-depleted condition. Dots represent the mean values for each experiment ($N = 3$) and the bar represents the average of the three values. *$P = 0.0142$ by Student's $t$ test. (F) Dot plot showing individual values for mean intensity of MTs at plus ends attached to aligned KTs in control and CKAP5-depleted cells ($n$ ~500 KTs) from three independent experiments. ****$P < 0.0001$ by Mann–Whitney test. (G) Dot plot showing individual values of mean MT plus end intensity of aligned KT-attached MTs in HeLa Kyoto un-transfected, cells expressing CKAP5 esiRNA and CKAP5$^{853-2032}$-GFP expressing cells in CKAP5-depleted background ($n$ ~400 KTs) from three independent experiments. ****$P < 0.0001$ by one-way ANOVA. (H) Representative immunofluorescence image showing CENP-E (green) localization at aligned KTs of cells treated with 10 μM paclitaxel for 30 min compared to that of DMSO-treated cells. ACA (red) was stained as KT marker. Scale bar = 5 μm. (I) Dot plot showing mean intensity of endogenous CENP-E at aligned KTs normalized with that of ACA where each dot represents individual KT ($n$ ~900) quantified from multiple cells from three independent experiments. The difference was shown to be nonsignificant by the Student's $t$ test. (J) Representative immunofluorescence image showing CENP-E (green) localization at aligned KTs of CKAP5-depleted cells treated with 3.3 μM nocodazole or DMSO for 10 min compared to that of control metaphase cells (region marked in box). ACA (red) was stained as KT marker. Scale bar = 5 μm. (K) Dot plot showing mean intensity of endogenous CENP-E at aligned KTs (highlighted with boxes) normalized with that of ACA where each dot represents individual KT ($n$ ~650) quantified from multiple cells from three independent experiments. ****$P < 0.0001$ by one-way ANOVA (L) Representative immunofluorescence image showing CENP-E (gray) localization at aligned KTs (region shown in boxes) of cells expressing CKAP5$^{853-2032}$-WT-GFP vs CKAP5$^{853-2032}$-KK/AA-GFP. Knl1 (red) was used as KT marker. Scale bar = 5 μm. (M) Dot plot showing mean intensity of CENP-E at aligned KTs (shown in boxes in (L)) normalized with that of Knl1 where each dot represents individual KT ($n$ ~900) quantified from three independent experiments. ****$P < 0.0001$ by Mann–Whitney test. All data represent mean $+/-$ SEM. Source data are available online for this figure.

considering that larger angles destabilize the kMT–KT attachment (Fig. 5F) (Renda et al, 2022), and the ratio of the number of MTs attached from the same pole to the number of MTs from the opposite pole (Lampson and Grishchuk, 2017). Each of these factors is thought to promote chromosomal bi-orientation by stabilizing kMTs that are more closely aligned to the KT–KT axis (Fig. 5E,F). Because of inter-KT tension stabilizing a pair of amphitelic sister KTs, no kMT detachment occurs in this case and new microtubule attachment to the KT pair is ignored. More details on the computational models are included in "Methods", and the model parameters are noted in Table EV1.

The simulation was carried out for ~ 20 min, which is sufficiently long for the KTs in a control cell to achieve bi-orientation and congress to the metaphase plate (Paul et al, 2009). Our results revealed that significant erroneous attachments persisted in the CKAP5-depleted condition, while mostly the amphitelic attachments were formed in the control cells (Fig. 5G). CKAP5 depletion leads to a significant percentage of chromosomes with syntelic attachments (~33%) remaining after 20 min of simulation time with respect to a small percentage of merotelic attachments (~0.56%). Merotelic attachments were not observed in the control cells; however, a very small percentage of syntelic (~0.07%) and monotelic or unattached (~0.4%) chromosomes remained (Fig. 5G). The end-on attached sister KTs in the control cells were found near the spindle mid-zone that oscillates around it (Fig. 5H,I; Movie EV1). In the CKAP5-depleted cells, while the amphitelic KTs congressed mostly near the spindle center, syntelic attachments remained close to the poles and often formed behind the centrosomes (Fig. 5J,K; Movie EV2). Contrary to the control scenario, we found sister KTs with amphitelic attachments had a larger spread of the distribution away from the spindle center in the CKAP5-depleted cells (Fig. 5H–K). Interestingly, our simulation revealed that a relatively larger number of kMTs are associated with KTs in the CKAP5-depleted condition compared to the control scenario (Fig. 5L). We further noticed that varying the severity of CKAP5 depletion by reducing the force strengths of kMT–KT attachments yielded reduced amphitelic attachments and increased syntelic attachments (Fig. 5M). Overall, our computational study qualitatively captures the defects in

chromosome distribution patterns and the occurrence of the erroneous attachments under the CKAP5-depleted condition as displayed in the experimental data (Figs. 3 and EV2E,F).

## Sensitivity to CKAP5 perturbation is associated with a high degree of aneuploidy and with sensitivity to CENP-E depletion

Having established the importance of CKAP5 activity for recruiting CENP-E to KTs, and the functional consequences of its inhibition, we sought to explore whether CENP-E expression and essentiality were associated with chromosome instability and with those of CENP-E in human cancer. For this, we analyzed mRNA and protein expression levels of 1408 and 339 human cancer cell lines, respectively, as well as CRISPR screens for 1078 human cancer cell lines, and associated them with the cells' ploidy status, using data obtained from the Dependency Map database (Cohen-Sharir et al, 2021; Dempster et al, 2021; Nusinow et al, 2020; Tsherniak et al, 2017). The mRNA and protein expression levels of CKAP5 were similar between chromosomally stable, near-diploid cell lines and chromosomally unstable, highly aneuploid cell lines (Fig. 6A,B). However, the highly aneuploid cells were much more sensitive to the CRISPR-mediated knockout of CKAP5 (Fig. 6C,D), confirming a functional link between CKAP5 and chromosomal instability. Similar to other kinetochore-related genes (Fig. EV4), both expression and dependency levels were significantly correlated between CKAP5 and CENP-E (Fig. 6E–G), providing further support for their functional interaction. Taken together, these results suggest that the functional link between CKAP5 and CENP-E is ubiquitous in cancer, and that highly aneuploid, chromosomally unstable cancers might be more sensitive to CKAP5 depletion.

## Discussion

Earlier studies in human cells showed that CKAP5 knockdown leads to the hyperstabilization of microtubules attached to KTs of polar chromosomes (Herman et al, 2020). Our detailed analysis

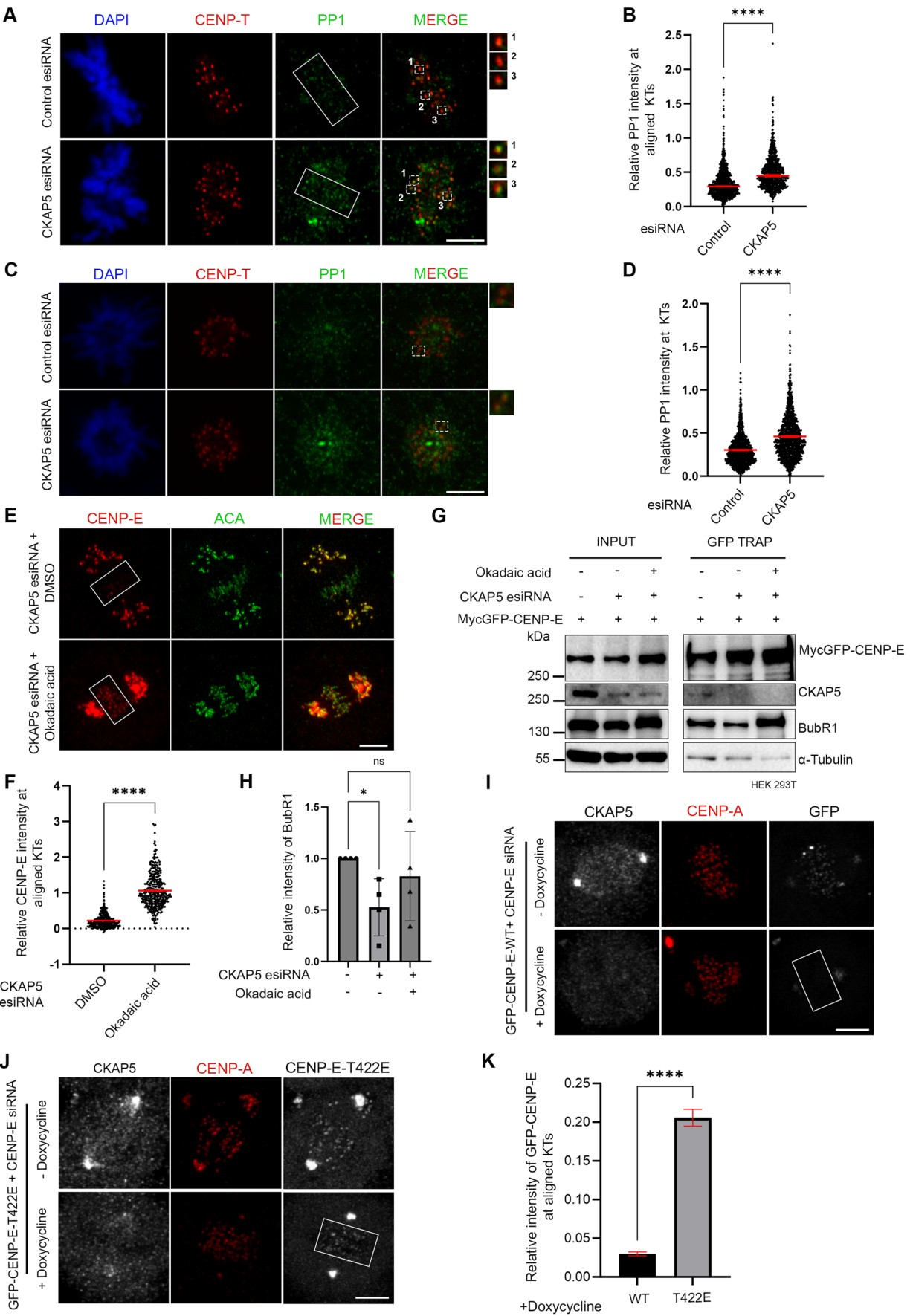

**Figure 4. CKAP5 regulates PP1 recruitment to KTs.**

(A) Immunofluorescence images (single plane) showing mock-depleted and CKAP5-depleted HeLa Kyoto cells stained for PP1 (green), CENP-T (red), DAPI (blue). Dashed boxes show a few aligned KTs -associated PP1 and enlarged view of the same are shown in insets. (B) Dot plot showing individual KT mean intensity ratio of PP1/CENP-T at aligned kinetochores (region marked in rectangular boxes in (A)). Mean intensity values from three independent experiments are plotted ($n$ ~1000 KTs). ****$P < 0.0001$ by Mann–Whitney test. (C) Immunofluorescence images (single plane) showing mock-depleted and CKAP5-depleted HeLa Kyoto cells that are arrested in prometaphase stained for PP1 (green), CENP-T (red), DAPI (blue). Dashed boxes show a few prometaphase KTs -associated PP1 and enlarged view of the same are shown in insets. (D) Dot plot showing individual KT mean intensity ratio of PP1/CENP-T at prometaphase KTs ($n$ ~1400 KTs). Mean intensity values from three independent experiments are plotted. ****$P < 0.0001$ by Mann–Whitney test. (E) Representative immunofluorescence images showing CENP-E (red) localization at KTs in CKAP5-depleted cells treated with DMSO or 0.25 μM Okadaic Acid (OA) for 1 h. Rescue of CENP-E to the partially aligned KTs in the presence of OA is shown in box. ACA (green) was used as KT marker. (F) Plot of CENP-E fluorescence intensity normalized to ACA at aligned KTs in the CKAP5 depletion and CKAP5 depletion plus OA conditions. Data shown are mean values of three independent experiments, where intensities of individual KTs ($n$ ~370 KTs) from the maximum projection images were quantified. ****$P < 0.0001$ by Mann–Whitney test. (G) GFP Trap pulldown showing CENP-E-BubR1 interaction in the absence and presence of 0.25 μM OA in endogenous CKAP5-depleted cells expressed with MycGFP-CENP-E. (H) Graph showing mean intensity of coprecipitated BubR1 normalized with MycGFP-CENP-E pull-down bands from four independent experiments of G. *$P = 0.0332$ by one-way ANOVA. (I, J) Representative immunofluorescence images showing KT localization of GFP-CENP-E-WT (I) and GFP-CENP-E-T422E (J) in the presence or absence of doxycycline in doxycycline-inducible CKAP5 KO HeLa cells under endogenous CENP-E depleted background. CENP-A (red) was used as KT marker. (K) Plot showing the significant rescue of CENP-E-T422E when compared to CENP-E-WT at aligned KTs (region marked with box in (I, J)) of CKAP5 KO cells. Mean intensities of GFP-CENP-E-WT and T422E mutant at aligned KTs normalized with that of CENP-A quantified from individual KTs ($n$~650–900 KTs) of multiple cells from three independent experiments are plotted. ****$P < 0.0001$ by Mann–Whitney test. Scale bar of all images = 5 μm. All data represent mean $+/-$ SEM. Data Information: Rectangular boxes represent region (aligned KTs) taken for PP1 intensity analysis. Dotted boxes in merged image shows regions (single KTs of aligned chromosomes) that are enlarged in insets which shows increase in KT intensity of PP1 (green) in CKAP5-depleted condition. Each number in insets corresponds to the respective numbered dotted box (A). Dotted box in merged image shows few prometaphase KTs that are shown enlarged in insets. PP1 levels at KTs (green) are increased in CKAP5-depleted prometaphase cells (C). Rectangular boxes show CENP-E levels at the aligned KTs that are considered for intensity analysis. CENP-E levels (red) showed rescue upon treatment with okadaic acid (E). Rectangular boxes show region (aligned KTs) taken for GFP-CENP-E intensity analysis. GFP-CENP-E T422E showed stabilization when compared to GFP-CENP-E-WT (GFP in gray) under doxycycline-induced CKAP5 depletion (I, J). Source data are available online for this figure.

specifically of the cold stable kMTs also showed similar hyperstabilization in the CKAP5-depleted cells (Fig. 3). The contribution of hyper-stable microtubules in promoting KT attachment errors has earlier been reported (DeLuca et al, 2011; Vallot et al, 2018). CKAP5 has been shown to display MT plus end destabilizing activity, although it can promote MT polymerization (Brouhard et al, 2008; Miller et al, 2016; Shirasu-Hiza et al, 2003; van Breugel et al, 2003). Therefore, MT hyperstabilization in CKAP5-depleted cells could also contribute to the attachment errors that we have observed. This effect could also be associated with the increased recruitment of PP1 to the KTs or vice versa (Fig. 4). Timely recruitment of PP1 to the KTs is critical for both congression of pole-proximal chromosomes to the metaphase plate and the subsequent stabilization of KT–MT attachment (Conti et al, 2019; Kim et al, 2010; Liu et al, 2010; Redli et al, 2016; Sivakumar et al, 2016; Trinkle-Mulcahy et al, 2003). Specifically, after congression, PP1 dephosphorylates proteins at the outer KTs allowing stable KT–MT attachment and SAC silencing (Emanuele et al, 2008; Kim et al, 2010; Liu et al, 2010; Nijenhuis et al, 2014). Here, we find that abrogation of a microtubule plus end-associated protein CKAP5, leads to untimely recruitment of PP1 to the KTs early from the prometaphase stage (Fig. 4). In the CKAP5-depleted cells, though many of the chromosomes are localized near the spindle mid-zone, they could align only partially (Fig. 3) indicating that there are problems in correct attachment to be established. It is possible that the recruitment of PP1 at a premature stage such as prometaphase, promotes erroneous attachments to stabilize. Consistently, our results showed that merotelic and syntelic attachments are induced in CKAP5-depleted condition (Figs. 3, 5 and EV2). How could PP1 recruitment be stimulated in the absence of CKAP5? Since CKAP5 depletion leads to increased density of MTs, a possibility is that the presence of CKAP5 on the microtubules controls the rate of kMT stabilization and associated PP1 recruitment to the KTs. CKAP5 serves a direct role in this process since we showed that expression of CKAP5 C-terminus containing TOG4 and TOG5

(853–2032) rescues the increased PP1 localization as well as MT hyperstabilization (Figs. EV2, EV3, and Fig. 3).

How could the increased microtubule stability and untimely PP1 recruitment induce CENP-E removal from the KTs? Earlier studies indicated that CENP-E removal from KTs is facilitated in cells treated simultaneously with Aurora B inhibitor and Paclitaxel, a MT stabilizing drug, suggesting that optimum MT stability and Aurora B-mediated phosphorylation has a role in stabilizing CENP-E levels at KTs (Gurden et al, 2018). Interestingly, mimicking the CKAP5 depletion-induced MT hyperstability phenotype using paclitaxel in control cells did not cause CENP-E removal from the KTs (Fig. 3). This is presumably due to the Aurora B activity at KTs as well as inhibition of dynein-mediated cargo movement or stripping of KT proteins through paclitaxel-stabilized MTs as reported earlier (Forer et al, 2018; Gurden et al, 2018). Therefore, it is possible that hyper-stabilized MTs induced by CKAP5 depletion could lead to defects in KT–MT attachments and under such condition, PP1-mediated delocalization of CENP-E is induced. Furthermore, we have also shown that loss of CENP-E in CKAP5-depleted cells was significantly rescued upon a brief nocodazole treatment, suggesting that CKAP5 is critical for maintaining optimum MT stability and PP1 localization and thereby, controlling CENP-E level at aligned KTs. This is also supported by our data that the CKAP5$^{853-2032}$-KK/AA mutant that induces MT hyperstability also leads to a significant loss of CENP-E from the partially aligned KTs (Fig. 3). A recent report also suggested that the dephosphorylated form of CENP-E specific to the Aurora B-targeting site (T422A) cannot localize to aligned KTs due to increased stripping by dynein (Eibes et al, 2023). Consistently, we have shown that CENP-E removal from the KTs in CKAP5-depleted cells requires MTs, and the level of PP1 at the KTs is upregulated under that condition. Further, the phospho-mimicking CENP-E mutant (T422E) is substantially stabilized at KTs in the CKAP5-depleted cells (Fig. 4) and depletion of Spindly together with CKAP5 rescues KT level of endogenous CENP-E

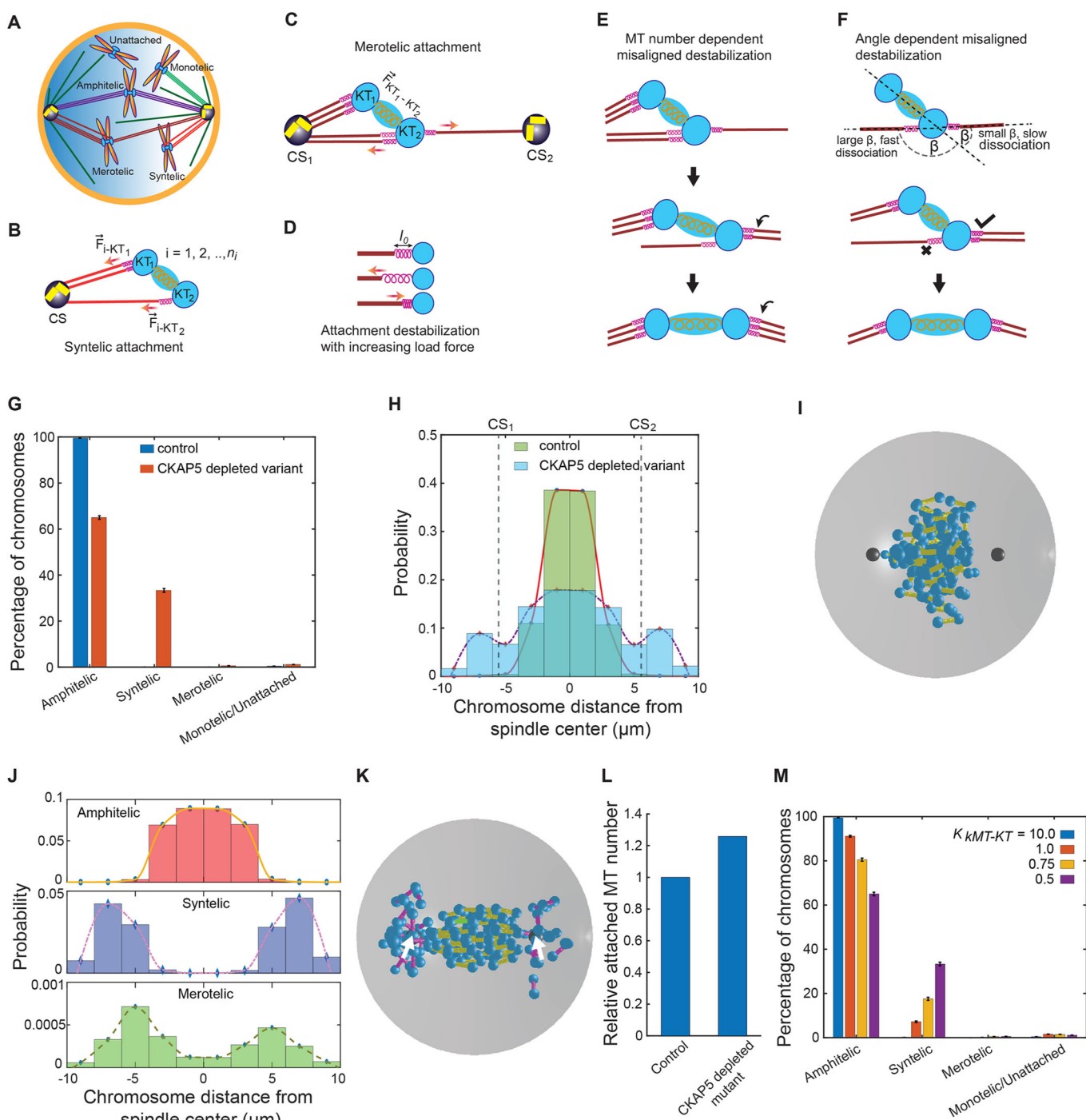

(Fig. EV1). Altogether, the results support the possibility of increased dynein-mediated stripping of presumably, the dephosphorylated form of CENP-E from the partially aligned KTs in the absence of CKAP5.

CENP-E in its phosphorylated form is known to be stabilized at KTs (Huang et al, 2019; Liao et al, 1994). Specifically, BubR1 interacts with and phosphorylates CENP-E at its C-terminus and thereby stabilizes it at the KTs, which is required for establishing stable end-on attachment (Chan et al, 1998; Huang et al, 2019; Johnson et al, 2004;

Legal et al, 2020; Mao et al, 2003). Our results show that CKAP5 depletion impairs CENP-E-BubR1 interaction leading to CENP-E removal from the partially aligned KTs and this interaction was rescued upon expression of CKAP5^853–2032 and also in PP1 inhibited condition (Figs. 1, 2, and 4). Given that CKAP5^853–2032 rescues PP1 levels at KTs and also rescues CENP-E-BubR1 interaction in CKAP5-depleted cells, the above results also support a direct role of CKAP5 in CENP-E stabilization by maintaining optimum levels of PP1 at the MT-attached KTs. Therefore, KT-attached MTs of optimal stability

**Figure 5.  Results of the computational model with error correction mechanism.**

(A) Schematic of various types of KT–MT attachments. (B–F) Model depicting the mechanism of correction of syntelic and merotelic attachments; (B, D) shows the force-dependent correction of syntelic attachments and (C–F) show the mechanism of correction of merotelic attachments through force dependence, attached MT number dependence and MT-KT attachment angle dependence. (G) Shows the percentage of chromosomes with various types of attachments in control and CKAP5-depleted mutant HeLa cells in our in silico model. $n = 2000 \times 80$ KT pairs. (H) Chromosome distribution in control and mutant HeLa cells after 20 min from nuclear envelope breakdown and the onset of "search-and-capture" process; two vertical dashed lines denote the position of the centrosomes along the spindle axis. (I) Snapshot of control cell after 20 min from the onset of nuclear envelope breakdown. Cyan spheres indicate attached and unattached KTs; yellow, pink, and green cylinders indicate amphitelic, syntelic, and merotelic chromosomes, respectively; black spheres indicate centrosomes. (J) Distribution of amphitelic, syntelic and merotelic chromosomes in CKAP5-depleted mutant. $n = 2000 \times 80$ KT pairs (K) Snapshot of a mutant HeLa cell 20 min after nuclear envelope breakdown; color scheme is the same as the sub-figure (I). Centrosomes are placed at ( ± 6.0, 0.0, 0.0) and denoted by two white arrows. (L) The relative number of total MTs connected to the KTs with amphitelic attachments in control and CKAP5-depleted mutant. (M) Variation of capture statistics with kMT–KT force strength. All the simulations were done with systems of 80 chromosomes and averages were taken over 2000 systems for every parameter value. $n = 2000 \times 80$ KT pairs. The error bars in (G, M) represent the standard error of mean measured with respect to the corresponding mean values of the data. Data Information: Simulations were done with systems of 80 chromosomes (80 KT pairs) with 2000 different initial configurations of the chromosomes; $n = 2000 \times 80$ KT pairs. The bar graphs in (G, M) are presented as mean ± SEM. Source data are available online for this figure.

mediated by CKAP5 could play a critical role in maintaining optimal PP1 levels so as the CENP-E levels at KTs.

Our computational model reveals that weakening of the kMT–KT interaction forces leads to error-prone phenotypes in the presence of a kMT–KT force-dependent error correction mechanism. We presume that kMT–KT spring-like attachments, mimicking the mechanical interaction with the KT, detach when experiencing a high load force (Klumpp and Lipowsky, 2005; Kunwar et al, 2011). The inability of CKAP5-depleted cells to stabilize CENP-E at the KTs is believed to weaken the kMT -KT coupling, thus reducing the load force on the kMT–KT spring-like attachments and prematurely stabilizing them with error-prone phenotypes. Our model predicts that CKAP5-depleted cells evolve with a higher percentage (~33%) of syntelic chromosomes due to the impaired error correction mechanism. In addition, syntelic chromosomes lack the necessary bi-orientation force as they are attached to only one mitotic spindle pole and therefore have a tendency to remain close to the poles. Since pulling force on KTs due to depolymerization of kMTs from opposite spindle poles is essential for the positioning and proper alignment of chromosomes at the metaphase plate (Maiato et al, 2017), syntelic chromosomes fail to congress to the metaphase plate as they experience MT depolymerization force from only one spindle pole. Uncaptured and monotelic chromosomes dispersed behind the poles have a higher chance of becoming syntelic; whereas chromosomes that wander near the metaphase plate can be captured from both the spindle poles. Consequently, the scattered distribution of syntelic chromosomes appears primarily behind the poles (Fig. 5K), which is qualitatively similar to our experimental data (Fig. EV2E,F). Thicker kMTs near the metaphase plate in CKAP5-depleted cells observed in our experiments (Fig. 3A,F) could arise from the slow turnover of the kMT–KT bond, leading to a relatively larger number of kMT attachments to the amphitelic chromosomes (Fig. 5L). Overall, our numerical results concur with the experimental outcome of reduced localization of CENP-E due to CKAP5 depletion. While our simplified numerical approach with regulated kMT–KT force interactions qualitatively reproduced the observed erroneous phenotypes in the CKAP5-depleted condition, direct measurement of these forces is beyond our experimental scope. This necessitates additional experiments specifically designed to measure the tension applied to the kMT–KT interface through CKAP5-mediated KT stabilized CENP-E, validating our simulation-based hypothesis and offering valuable insights into the mechanical aspects of kinetochore–microtubule interactions. The present model also ignores the explicit dynamics of individual motor proteins and their active interactions with kMTs. Temporal regulation

of phosphorylation-dephosphorylation of KT proteins is a major driver of error-free chromosome segregation in mitosis. The results of this work demonstrate that specific microtubule-associated protein (MAP), CKAP5 orchestrates this process by controlling the stability of KT-attached MTs, deregulation of which facilitates PP1 recruitment and stabilizes erroneous attachments. Since MT plus end stability is regulated by other MAPs as well, their co-operating roles along with CKAP5 should be explored in the future.

Our human cancer cell line data analysis suggests that the interaction between CKAP5 and mitotic errors is bidirectional—not only does CKAP5 depletion lead to mitotic errors but chromosomally unstable cells are also more sensitive to CKAP5 depletion (Fig. 6). These findings are consistent with a recent report that found that CKAP5 silencing could lead to lethality of chromosomally unstable cancer cells (Chatterjee et al, 2023). It is also in line with previous findings that highly aneuploid cells are more sensitive to the depletion of specific components of the SAC and its regulators (Cohen-Sharir et al, 2021; Marquis et al, 2021; Quinton et al, 2021), due to the inability of such cells to cope with the severe mitotic errors that SAC perturbation could cause. Importantly, the cellular sensitivity to CKAP5 depletion was significantly correlated with the sensitivity to CENP-E depletion, consistent with their functional relationship (Fig. 6). Overall, the cell line analysis supports the generalizability of our findings, and strengthens the notion that CKAP5 might be a synthetic lethality of chromosomally unstable, aneuploid cancer cells.

## Methods

### Reagents and antibodies

Dulbecco's modified Eagle's medium (DMEM) and fetal bovine serum (FBS) were purchased from ThermoFisher Scientific, Massachusetts, USA. Penicillin-streptomycin solution was obtained from Himedia, Mumbai, India. Thymidine, Calbiochem Insolution Okadaic acid, Eg5 kinesin inhibitor Dimethylenastron (DMA), EGTA, PIPES, HEPES, MG132, and DAPI were purchased from Sigma-Aldrich, St Louis, MO. In total, 16% PFA was obtained from Electron Microscopy Sciences, PA, USA. Transfection reagents Lipofectamine 3000 and RNAiMax and ProLong Gold antifade mountant were obtained from ThermoFisher Scientific, MA, USA. Transfection reagent PEI (Polyethyleneimine) was obtained from Polysciences, PA, USA. MISSION® esiRNA targeting FLUC (Cat # EHUFLUC) against

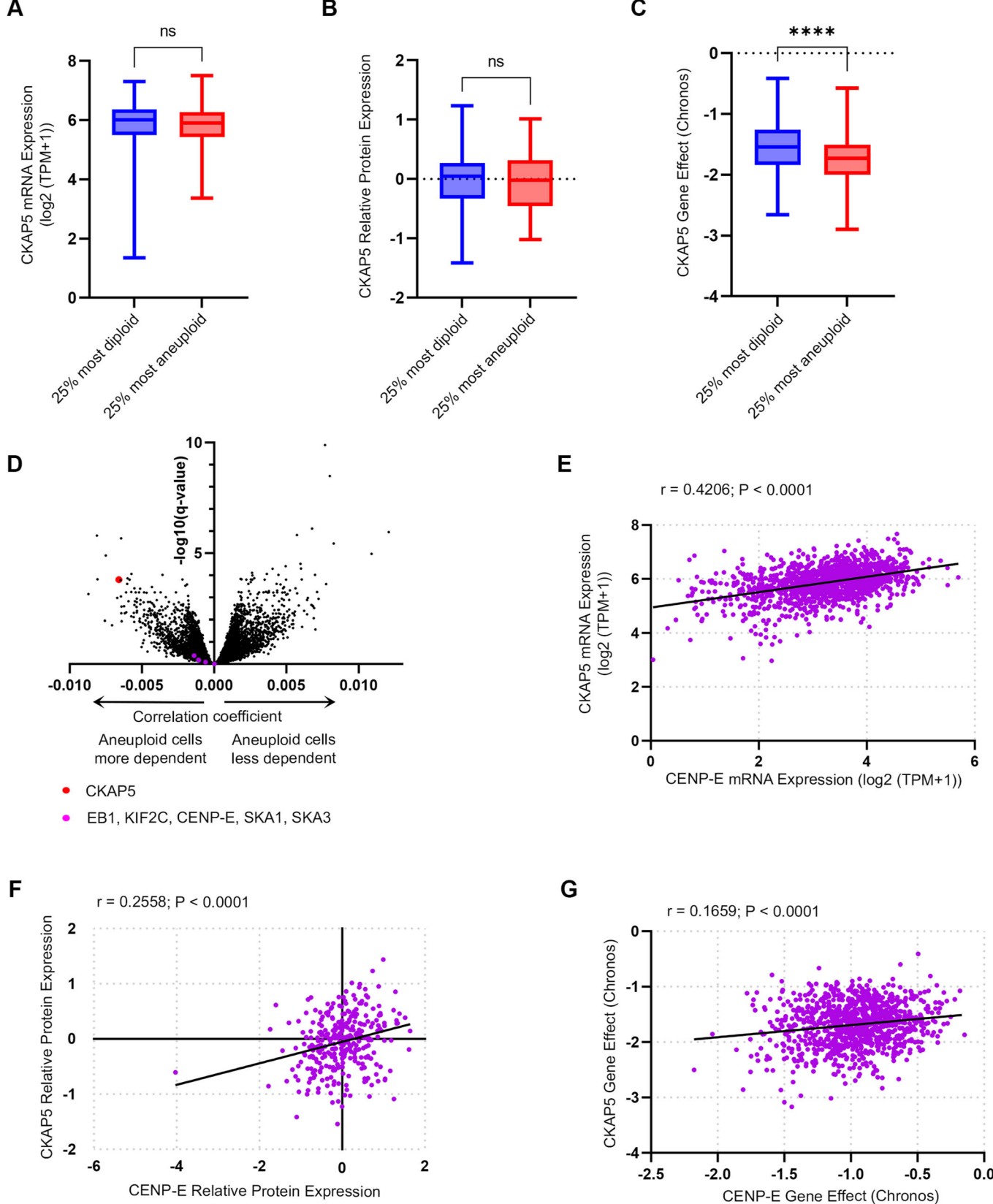

**Figure 6. Sensitivity to CKAP5 perturbation is associated with a high degree of aneuploidy and with sensitivity to CENP-E depletion.**

Analyses of data from hundreds of cancer cell lines obtained from the cancer-dependency map release 22Q4 (Data ref: Tsherniak et al, 2017). (**A, B**) Chromosomally stable, near-diploid cell lines and chromosomally unstable, highly aneuploid cell lines express similar levels of CKAP5 on the mRNA (**A**) and protein (**B**) levels. $n = 381$ and 341 for the bottom and top aneuploidy quartiles, respectively, in (**A**) and $n = 91$ and 102 for the bottom and top aneuploidy quartiles, respectively, in (**B**). Two-sided $T$ test (significance threshold: $P$ value < 0.05) (**A**) $P$ value = 0.8221, (**B**) $P$ value = 0.3615. (**C**) Highly aneuploid cells are significantly more sensitive to the CRISPR-mediated knockout of CKAP5. $n = 274$ and 284 for the bottom and top aneuploidy quartiles, respectively. Two-sided $T$ test, ****$P < 0.0001$. In all box plots, bar: median; box: 25th and 75th percentile; whiskers: minimal and maximal values. (**D**) Correlation between aneuploidy and sensitivity to CRISPR gene perturbation. Sensitivity to CKAP5 perturbation is very strongly associated with aneuploidy, while sensitivity to perturbation of other spindle/kinetochore genes is not correlated to aneuploidy at all. (**E, F**) CKAP5 and CENP-E are co-expressed on the mRNA (**E**) and protein (**F**) levels, across cancer cell lines. Pearson correlation; (**E**) $r = 0.4206$, $P$ value < 0.0001, (**F**) $r = 0.2558$, $P$ value < 0.0001. (**G**) Sensitivity to CKAP5 perturbation by CRISPR is associated with sensitivity to CENP-E depletion by CRISPR across cancer cell lines. Pearson correlation; $r = 0.1659$, $P$ value < 0.0001. The data in plots (**E–G**) is normalized to the mRNA expression of the proliferation marker MKI67 as a covariate using a linear model. Source data are available online for this figure.

Firefly Luciferase and MISSION® esiRNA against CKAP5 (Cat # EHU078221) were obtained from Sigma-Aldrich. GFP trap agarose beads (Cat # gta-20) for GFP pulldown were purchased from Chromotek, Munich, Germany. 3' UTR siRNA against CENP-E (5′-CCACUAGAGUUGAAAGAUA-3′), custom siRNA against Spindly (5′-GAAAGGGUCUCAAACUGAA-3′) and CKAP5 (5′-CAUG-CUCCACAGCAAACUCUC-3′) were purchased from GE Health care Dharmacon Inc. Rabbit polyclonal antibodies against CKAP5 (Cat # ab-86073), CENP-T (Cat # ab-220280), Knl1 (Cat # ab222055) were obtained from Abcam, Cambridge, MA, USA. Mouse monoclonal antibody against CENP-E (Cat # ab5093) and Bub1 (Cat # ab54893) were purchased from Abcam (MA, USA). Mouse monoclonal antibody against CENP-A (Cat # GTX13939) was obtained from GeneTex, CA, USA. Mouse monoclonal antibody against BubR1 (Cat # 612503) was obtained from BD Biosciences (California, USA). Mouse monoclonal antibodies for CENP-E (Cat # sc-376685), PP1 (Cat # sc-7482), GFP (Cat # sc-9996), C-Myc (Cat # Sc-40) were obtained from Santa Cruz Biotechnologies, CA, USA. Mouse monoclonal antibody against GFP (Cat # 632381) was purchased from Takara Bio Clontech, USA. Rabbit polyclonal antibody against pHecS55 (Cat # GTX70017) was obtained from GeneTex, CA, USA. Anti-centromere protein antibody, ACA (Cat # 15-234) was obtained from Antibodies Incorporated, CA, USA. Mouse monoclonal antibody against α-Tubulin (Cat # T6199) was obtained from Sigma-Aldrich, St Louis, MO. All HRP-conjugated secondary antibodies, Alexa 488, Alexa 647, and cy5-conjugated secondary antibodies were purchased from Jackson ImmunoResearch, PA, USA. Alexa 568-conjugated anti-rabbit secondary antibody was obtained from ThermoFisher Scientific, MA, USA. The dilutions of the primary antibodies used were: CENP-E (IF-1:200; WB-1:200), CENP-A (1:400), CENP-T (1:200), Knl1 (1:200), PP1 (IF-1:100; WB-1:500), ACA (1:700), α-Tubulin (IF-1:700; WB-1:2000), CKAP5 (IF-1:200; WB-1:2000), BubR1 (IF-1:200; WB-1:500), Bub1 (1:300), pHecS55 (1:200), GFP (WB-1:2000), and GST (WB-1:2000). Secondary antibodies were used at dilutions 1:1000 (IF) and 1:2000 (WB).

## Plasmids

All truncated and mutated CKAP5 (ch-TOG) constructs (CKAP5[1-1428], CKAP5[853-2032], CKAP5 [853-2032]-KK/AA, CKAP5[1429-2032]) were amplified from pBrain-ch-TOG KDP-GFP-shch-TOG (a kind gift from Stephen Royle, Addgene plasmid # 69113) and cloned to the same vector. The plasmid possesses shRNA against endogenous CKAP5 to which the CKAP5 cDNA is resistant. pCDNA3 eGFP empty vector (a gift from Doug Golenbock, Addgene plasmid # 13031) was used as empty GFP control. pCDNA5-FRT-TO-Myc-Lap-hCENP-E-WT, T422E and T422A mutants were obtained from Don W. Cleveland Lab, Department of Cellular and Molecular Medicine, University of California San Diego.

## Cell culture and transfection

HeLa Kyoto cells were obtained from Sachin Kotak, Indian Institute of Science (IISc), Bangalore (originally provided by Daniel Gerlich, IMBA, Vienna). HEK 293T and U2OS cells were originally obtained from ATCC. CKAP5 conditional knockout HeLa cell line (cKMKO C5.2) was a kind gift from Ian M Cheeseman (Whitehead Institute, MIT, USA). The cell lines were maintained in DMEM high glucose media supplemented with 10% FBS at 37 °C in a humidified atmosphere with 5% $CO_2$. For conditional knockout cell lines, DMEM high glucose media supplemented with 10% Tet-free FBS was used. Cells were incubated in 2 μM doxycycline for 72 h to get effective CKAP5 knockout. For transient transfection of all plasmids to HeLa Kyoto cells and 293T cells, Lipofectamine 3000 or PEI (Polyethyleneimine) reagents were used. For mock and CKAP5 depletion, Luciferase esiRNA and CKAP5 esiRNA were transfected using Lipofectamine RNAiMax reagent. To ensure that the cells are in their first mitotic division after gene knockdown, single thymidine-arrested cells were released and transfected with esiRNA after 6 h of release, followed by a second thymidine arrest for 24 h and fixed at 9th hour after release. For pull-down experiments, cells were synchronized at prometaphase by double thymidine block followed by thymidine release to DMEM containing DMA (Dimethylenastron, Eg5 kinesin inhibitor) for 12 h. For metaphase synchronization, double thymidine blocked cells were either released into DMEM containing DMA for 12 h followed by DMA washout with fresh DMEM containing MG132 for 2 h or simply thymidine release followed by MG132 treatment at 8th hour after release for 2 h (HEK 293T cells). Drugs were used at the following final concentrations: Thymidine, 2 mM; DMA, 5 μM; MG132, 25 μM; Okadaic acid, 0.25 μM/0.12 μM; Reversine, 250 nM; Nocodazole, 3.33 μM; Doxycycline, 2 μM; Paclitaxel, 10 μM.

## Immunofluorescence confocal microscopy and image analysis

Cells were pre-extracted for 5 min using 0.1% Triton X 100 in PHEM buffer (60 mM PIPES (pH 6.8), 25 mM HEPES (pH 6.9), 10 mM EGTA and 2 mM $MgCl_2$) before fixation in some cases. Cells were then fixed using 4% PFA in PHEM for 15 min at room temperature, followed by

blocking with 1% PBSAT (1× PBS, 1% BSA, 0.5% Triton x 100). For the cold stability assay, cells were incubated in ice for 5 min and fixed in ice-cold methanol for 20 min. For all CENP-E staining, cells were fixed in ice-cold methanol. Primary antibody incubations were done for 3 h or overnight at room temperature in a dark humidified chamber. Secondary antibody incubation was done for 1 h at room temperature. DNA was stained using DAPI, and the cells were mounted with ProLong Gold (Invitrogen). Images were acquired in Zeiss LSM 880 confocal microscope, Leica SP5 upright confocal microscope, and Olympus FluoView3000 with a 63× oil-immersion objective. For better visibility, image contrast was adjusted manually by changing the LUT values. Raw confocal images (no contrast adjustments) were quantified for individual KT intensity across z or from maximum projection images using ZEN blue software or ImageJ Fiji. Background intensity was measured from multiple regions within the cell away from the KTs, and average background intensities were subtracted from the KT intensities of proteins of interest. All mean KT intensities of required proteins were normalized with that of a KT marker quantified similarly. For MT plus end intensity, a line with 5 μm width was drawn across kMTs at the plus end, and the maximum intensity value was obtained from plot profile in Fiji (Liu et al, 2010). All graphs were plotted and analyzed for statistical significance using GraphPad Prism 9.

## Structured illumination microscopy (SIM) imaging

Fluorescence microscopic images were captured by a structured illumination method using an inverted Zeiss ELYRA PS1 micro-scope. Four lasers have been used for excitation: 405 nm (50 mW), 488 nm (200 mW), 561 nm (200 mW), and 642 nm (150 mW). Imaging was performed using a Zeiss oil-immersion objective (alpha Plan-apochromat DIC 63×/1.40 Oil DIC M27; numerical aperture (NA) 1.40 oil (Fig. 1B) and alpha Plan-apochromat DIC 100×/1.46 Oil DIC M27; numerical aperture (NA) 1.46 oil (Figs. 1E and 3D). Fluorescence light was spectrally filtered with emission filters (MBS– 405 + EF BP 420–495/LP 750 for laser line 405, MBS– 488 + EF BP 495–570/LP 750 for laser line 488, MBS– 561 + EF BP 570–650/LP 750 for laser line 561 and MBS–642 + EF LP 655 for laser line 642) and imaged using a PCO edge sCMOS camera. The respective bright-field images were taken using LED light sources. Acquired images have been processed using ZEN 2.0 structured illumination software.

## Pull-down assays

Prometaphase or metaphase-arrested cells (HeLa Kyoto and HEK 293T) were lysed in mammalian cell lysis buffer (20 mM Tris, 50 mM NaCl, 1 mM EGTA, 1% Triton X 100 with 1× protease inhibitor cocktail). For all pull-down experiments, agarose beads were equilibrated in lysis buffer and blocked using 2% BSA in lysis buffer. For pulling down MycGFP-CENP-E and CKAP5-GFP the cell lysates were incubated with GFP nanobody-coated beads (GFP TRAP agarose) for 4 h at 4 °C. For CENP-E-BubR1 interaction rescue experiment, MycGFP-CENP-E was pulldown by incubating cell lysates with C-Myc antibody for 24 h at 4 °C followed by incubation with Protein A/G agarose for 2 h at 4 °C. The beads were then washed and processed for western blotting. The immunoblots were visualized in the ChemiDoc XRS+ Imaging System (Bio-Rad, Hercules, CA, USA). All protein band intensities were quantified using volume analysis tool in Image Lab software.

## Statistical analysis

Data are presented as mean +/− SEM. The normally distributed data were analyzed with modified Student's (Welch) $t$ test at the 99% confidence level. For skewed datasets, Mann–Whitney test was used for analysis. Wherever applicable, one-way ANOVA followed by Tukey's multiple comparison tests were performed. The data were plotted and analyzed using GraphPad Prism 9 software. The figures were organized using Adobe Photoshop and Adobe Illustrator.

## Computational modeling

The effects of CKAP5 depletion in HeLa cells were investigated in situ by a three-dimensional mechanistic model. Our model consisted of a spherical cell of radius $R_{cell}$ with two static centrosomes placed $d_{cent}$ distance away from the cell center on opposite sides. Initially, $N_{CH}$ chromosomes (CHs) are placed randomly throughout the cell. Dynamically unstable microtubules (MTs) grow and shrink throughout the cell with their minus end attached to the centrosomes and search for the sister kinetochores (KTs) by their dynamic plus ends (Kirschner and Mitchison, 1986; Paul et al, 2009). The chromosome arms experience a polar ejection force away from the poles due to the astral microtubule tips hitting on them (Antonio et al, 2000; Brouhard and Hunt, 2005; Ke et al, 2009; Levesque and Compton, 2001; Rieder and Salmon, 1994). The kinetochores are captured by the astral microtubules with a certain probability once the MT hits the KT. The depolymerization force of the MT on the KT generates poleward tension and is assumed to be a spring force (Powers et al, 2009; Wei et al, 2007). The sister KTs are assumed to be attached by cohesion, which is modeled as a spring force. The formation of a proper bipolar spindle and the congregation of the chromosomes at the metaphase plate requires all the KT–MT attachments to be amphitelic (sister KTs connected to opposite spindle poles; see Fig. 5A). Other types of wrong attachments also occur during the formation of the mitotic spindle, namely syntelic attachments where both sister KTs are attached to MTs from the same pole (Fig. 5A,B) and merotelic attachments where one KT is attached to MTs from opposite poles (Fig. 5A,C). Correction of the merotelic and syntelic attachments is hampered by CKAP5 depletion. CKAP5 depletion is known to prematurely stabilize MT-KT attachments (Herman et al, 2020). We assume that the detachment rate of a kinetochore-attached MT (kMT) is proportional to the load force on the MT and that depletion of CKAP5 leads to the reduction of kMT load forces. All these components are included in the model in the following manner:

### Polar ejection force
We assume that the number of microtubules a distance $x$ away from the poles is an exponentially decaying function of the distance $x$ and goes as: $N(x) \sim \exp(-x/L)$ (Ferenz et al, 2009; Sutradhar et al, 2015). The net polar ejection force on a chromosome arm is calculated as

$$F_{ejection}^i(x) = A_{pe} \exp(-x/L) \tag{1}$$

Where $A_{pe}$ is a constant denoting the maximum ejection force and $L$ is the average MT length.

## Kinetochore–microtubule force

Motor proteins such as dynein, CENP-E interact with the kMT and KT to generate a poleward force. At the kinetochore, fibrous corona proteins attach to the kMT tips. The force arising from the interaction of the fibrous corona proteins and the kMT can be poleward or away from the pole depending on whether the MT is depolymerizing or polymerizing, respectively (Fernandez et al, 2009; Gardner et al, 2005; Gorbsky et al, 1987; Hyman and Mitchison, 1990; McIntosh et al, 2008; Waters et al, 1996). In our model, we assume that the tip of a captured MT is connected to the KT center via a spring (Powers et al, 2009; Thomas et al, 2016; Wei et al, 2007). The force exerted on a KT due to $i$th MT is given by:

$$F_{i-KT} = K_{kMT-KT}(x - l_{kMT-KT}) \tag{2}$$

Here, $x$ is the distance of the MT tip from the KT center and $l_{kMT-KT}$ is the effective length of the attachment spring and $K_{kMT-KT}$ is its stiffness constant. The net force on a KT is the sum of forces due to all attached MTs to that KT. In the CKAP5-depleted mutant, we assume the value of $K_{kMT-KT}$ is significantly lower than in the control cell.

## Sister KT cohesion force

The sister KTs are assumed to be attached via a cohesion spring (Thomas et al, 2016). The cohesion spring generates forces between the sister KTs given by:

$$F_{KT_1-KT_2} = K_{cohesion}(|\boldsymbol{x}_1 - \boldsymbol{x}_2| - l_{cohesion}) \tag{3}$$

Here $\boldsymbol{x}_1$ and $\boldsymbol{x}_2$ are the position vectors of the two sister KTs and $l_{cohesion}$ is the rest length of the cohesion spring, $K_{cohesion}$ is the stiffness constant of the KT–KT cohesion spring-like attachment.

## Inter-chromosome repulsion

To ensure that the chromosomes do not overlap, a steric repulsive force, $A_{CH-CH}^{steric}$, is considered between the chromosomes that scale as the inverse square of the mutual distance between a pair of chromosomes (Chatterjee et al, 2020).

## Repulsion from the boundary

To avoid the collapse of the chromosomes on the cell boundary, a repulsive force from the boundary is applied on the chromosomes that decays exponentially as a function of the distance from the boundary.

$$F_{CH-cellboundary}^{repulsion} = A_{boundary} \exp(-x/L) \tag{4}$$

## Syntelic attachment correction

We consider that syntelic attachments do not stretch sister KTs enough to generate stabilizing effect (Tanaka et al, 2002). Stretched sister KTs have a stabilizing effect on kMTs due to the reduced phosphorylation of kMT attachments by Aurora B kinase (Wang et al, 2010; Wang et al, 2012). In the absence of stabilization, the kMT turnover is assumed to be proportional to the effective linkage between the kMT and KT. The kMT–KT attachment springs are expected to be increasingly prone to breaking as load force increases (Klumpp and Lipowsky, 2005; Kunwar et al, 2011)

favoring the detachment of associated kMTs. In the model, the detachment of a single $i^{th}$ MT from syntelic attachments is determined by a rate dependent on the MT– KT force as:

$$R_{syntelic}^i = R_{detach} \exp(F_{i-kt}) \tag{5}$$

## Merotelic attachment correction

In addition to the load-dependent kMT detachment, it is anticipated that the detachment of a single MT from merotelic attachments is proportionate to two additional factors: the angle of the attachment bond with the inter-KT axis (Renda et al, 2022) and the number of MTs connected from the poles (Lampson and Grishchuk, 2017); see Fig. 5C–F. The detachment rate of an MT is given by

$$R_{syntelic}^i = R_{detach} \left(\frac{n_i}{n_j}\right) \exp(F_{i-KT}) \exp(-\beta) \tag{6}$$

Here, $n_i$ is the number of microtubules connected to KT from the pole on which the considered MT is nucleated and $n_j$ denotes the number of MTs connected to the KT from the opposite pole. $\beta$ denotes the angle between the attached kMT and the KT–KT axis (Fig. 5F). Based on the number of kMTs, the error correction occurs as follows: In our model, merotelic attachments arise when an MT from the opposite pole catches one of the already formed syntelic attached KT. Increasing the number of MTs from the opposite pole is expected to stretch the inter-KT cohesion spring, stabilizing the kMTs to the KTs due to Aurora B kinase activity while destabilizing the MTs that were previously syntelically attached to that KT (Lampson and Grishchuk, 2017); see Fig. 5E for a schematic depiction. Similarly, the angle-dependent detachment factor enhances the stability of MTs that are more aligned to the KT–KT axis, hence promoting chromosomal bi-orientation (Renda et al, 2022); see Fig. 5F for a schematic depiction. Merotelic attachments have a high chance of converting to amphitelic attachments by detaching preferentially the MTs whose removal would increase the inter-KT tension (Edelmaier et al, 2020; Lampson and Grishchuk, 2017) (Fig. 5C–F).

## Microtubule dynamic instability

MTs grow with a velocity $v_g$ and shrink with a velocity $v_s$. Free MTs undergo rescue and catastrophe with frequency $f_r$ and $f_c$, respectively (Joglekar and Hunt, 2002). A depolymerizing MT experiencing a load force has a corrected rescue frequency to maintain the attachments of the kMT to KT (Thomas et al, 2016), given by:

$$f_r^{i^*} = 1 - \exp\left(-\frac{F_{i-KT}^{depol}}{f_s}\right) \tag{7}$$

Where $F_{i-KT}^{depol}$ is the force exerted on the $i$th microtubule in its depolymerization state and $f_s$ is the stall force of the MT. MTs also have a length-dependent catastrophe to ensure proper positioning of the KTs along the mitotic spindle (Sutradhar et al, 2015). Therefore, the catastrophe frequency of the MTs experiencing a load is corrected according to:

$$f_c^{i^*} = R_{cat} l_{mt}^i \tag{8}$$

$R_{cat}$ is a rate constant and $l_{mt}^{i}$ is the length of the loaded MT.

The coarse-grained model used in this work is based on the CKAP5 depletion-induced effect on the spring-like attachments between the kMT tip and the KT. The spatial distributions of kMT-associated CKAP5 and CENP-E proteins on the KT surface and the structural changes of fibrous corona upon MT interaction (Magidson et al, 2015) do not feature in the present model. In this context, dynein-driven remodeling of the fibrous corona structure by stripping the corona elements toward the MT minus end, following the formation of a stable kMT–KT attachment can alter the forces at the KT (Auckland et al, 2020). A refined molecular model including the aforementioned factors would be a useful upgradation for quantitative predictions.

## Cancer cell line data analysis

Human cancer cell line aneuploidy scores (AS), mRNA and protein expression datasets, and CRISPR-Cas-9-based gene dependency scores (determined by the "Chronos" algorithm) were obtained from DepMap release 22Q4 (https://depmap.org/portal/) (Cohen-Sharir et al, 2021; Dempster et al, 2021; Nusinow et al, 2020; Data ref: Tsherniak et al, 2017). Cell lines were split into two groups of semi-diploid and highly aneuploid cell lines, correlating to the top and bottom quartiles of aneuploidy scores. A two-sided Student's t test was used to compare gene expression and CRISPR dependency values between the groups. Statistical analyses were performed in GraphPad Prism 9.1. To remove the effects of proliferation, we fitted linear regression models to the correlations and used the R "partialize" function to compute the residuals without the effect of MKI67 mRNA expression, which were then plotted.

# Data availability

All relevant data are available. Simulation Code, written in C, can be accessed at https://github.com/PinakiNayak13/CKAP5-computational-data.git.

# Peer review information

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

## Acknowledgements

The authors thank Dr. Don Cleveland, UC San Diego for MycGFP-CENP-E-WT, T422A and T422E mutant plasmids. The authors also thank Dr. Iain Cheeseman, Whitehead Institute, MIT, USA for providing CRISPR-Cas-9-based CKAP5 inducible knockout HeLa cells. Financial supports from DBT, Government of India and DST-SERB, Government of India to TKM are thankfully acknowledged. Financial support from the European Research Council (grant #945674), the Israel Science Foundation (grant #1805/21) and the Israel Cancer Association (grant #20230018) to UBD are thankfully acknowledged. Dr. Ben-David receives research funding from NovoCure. PN acknowledges funding support from CSIR, India. AS and RP acknowledge funding support from IACS, Kolkata, India.

## Author contributions

**R Bhagya Lakshmi**: Conceptualization; Data curation; Formal analysis; Validation; Investigation; Visualization; Methodology; Writing—original draft; Writing—review and editing. **Pinaki Nayak**: Data curation; Formal analysis; Methodology; Writing—original draft; Writing—review and editing. **Linoy Raz**: Data curation; Formal analysis; Methodology; Writing—original draft; Writing—review and editing. **Apurba Sarkar**: Formal analysis; Methodology; Writing—original draft; Writing—review and editing. **Akshay Saroha**: Data curation; Methodology. **Pratibha Kumari**: Data curation; Methodology. **Vishnu M Nair**: Methodology. **Delvin P Kombarakkaran**: Methodology. **S Sajana**: Formal analysis. **Sanusha M G**: Formal analysis; Methodology. **Sarit S Agasti**: Methodology. **Raja Paul**: Conceptualization; Formal analysis; Supervision; Funding acquisition; Methodology; Writing—original draft; Writing—review and editing. **Uri Ben-David**: Conceptualization; Funding acquisition; Methodology; Writing—original draft; Writing—review and editing. **Tapas K Manna**: Conceptualization; Supervision; Funding acquisition; Visualization; Methodology; Writing—original draft; Project administration; Writing—review and editing.

## Disclosure and competing interests statement

The authors declare no competing interests.

# Expanded View Figures

**Figure EV1. CKAP5 depletion induces chromosome misalignment and removal of CENP-E from kinetochores.**

(A) Representative immunofluorescence images showing level of CKAP5 depletion and chromosome misalignment phenotype. (B) Quantification of chromosome misalignment phenotypes ($n > 100$ cells in each case) in CKAP5-depleted cells compared to mock-treated (control esiRNA) cells from three independent experiments. Data represents mean $+/-$ SEM. (C, D) Western blots showing level of CKAP5 depletion and corresponding cellular levels of CENP-E. (E) Representative immunofluorescence images showing removal of CENP-E from partially aligned KTs of inducible Cas-9 expressing CKAP5 knockout HeLa cells (regions shown in box). (F, G) Immunofluorescence images showing loss of CENP-E localization at the partially aligned metaphase KTs (regions show in box) and prometaphase KTs of U2OS cells depleted of endogenous CKAP5. (H) Representative immunofluorescence images showing rescue of CENP-E localization at aligned KTs in CKAP5-Spindly co-depleted HeLa Kyoto cells compared to control metaphase and CKAP5-depleted cells (regions shown in box). (I) Dot plot showing mean intensity of endogenous CENP-E at aligned KTs (highlighted with boxes) normalized with that of ACA where each dot represents individual KT ($n = 400$–$550$) quantified from multiple cells from three independent experiments. ****$P < 0.0001$ by one-way ANOVA. Data represents mean $+/-$ SEM. (J) Western blot showing levels of CKAP5 and Spindly depletion in HeLa Kyoto cells. (K) Western blot showing endogenous CKAP5 depletion levels in the cells expressing GFP-tagged CKAP5-FL vs. different deletion constructs. (L) Immunofluorescence images showing chromosome phenotype in CKAP5-depleted cells (HeLa Kyoto) and in cells expressing fusion construct consisting of CKAP5[853-2032]-GFP and CKAP5-shRNA. (M) Plot showing percentage of cells ($n = 73$ cells) with less than 10 polar chromosomes in CKAP5[853-2032]-GFP-CKAP5-shRNA cells compared to CKAP5 esiRNA-treated cells (HeLa Kyoto). (N) Representative immunofluorescence image of HeLa Kyoto cell expressing CKAP5[1-1429]-GFP in endogenous CKAP5-depleted background showing no recognizable KT or spindle localization. Scale bar of all images $= 5\,\mu m$. Data Information: Chromosome congression defect and level of CKAP5 depletion (red) by immunofluorescence imaging (A). Rectangular boxes show aligned KTs with normal CENP-E localization in control vs. reduced CENP-E localization in CKAP5-depleted conditions (E, F). Doxycycline-inducible CKAP5 KO HeLa cells in (E) and U2OS cells in (F). CKAP5-depleted cells shows reduced CENP-E levels (green) at KTs compared to control cells when arrested in prometaphase by Eg5 kinesin inhibition (G). Rectangular boxes show region (aligned KTs) considered for CENP-E intensity analysis. Rescue in CENP-E levels (green) are observed upon co-depletion of spindly and CKAP5 (H). DAPI staining of chromosomes showing reduction in number of polar chromosomes upon expression of CKAP5[853-2032]-GFP compared to that in CKAP5-depleted condition (L). CKAP5[1-1429]-GFP (green) showing no localization in HeLa Kyoto cells when compared to CKAP5-FL-GFP (green) under endogenous CKAP5-depleted condition (N).

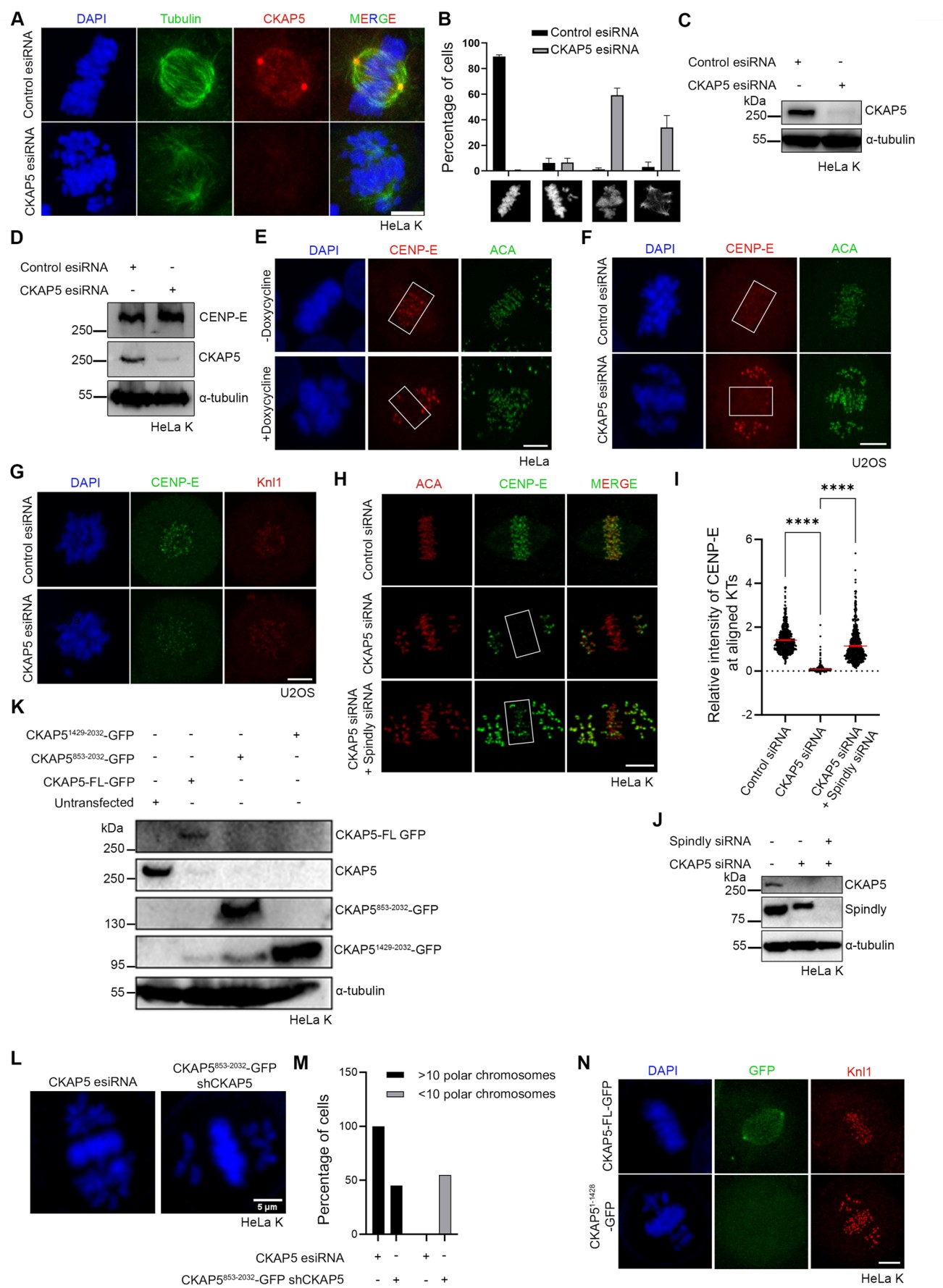

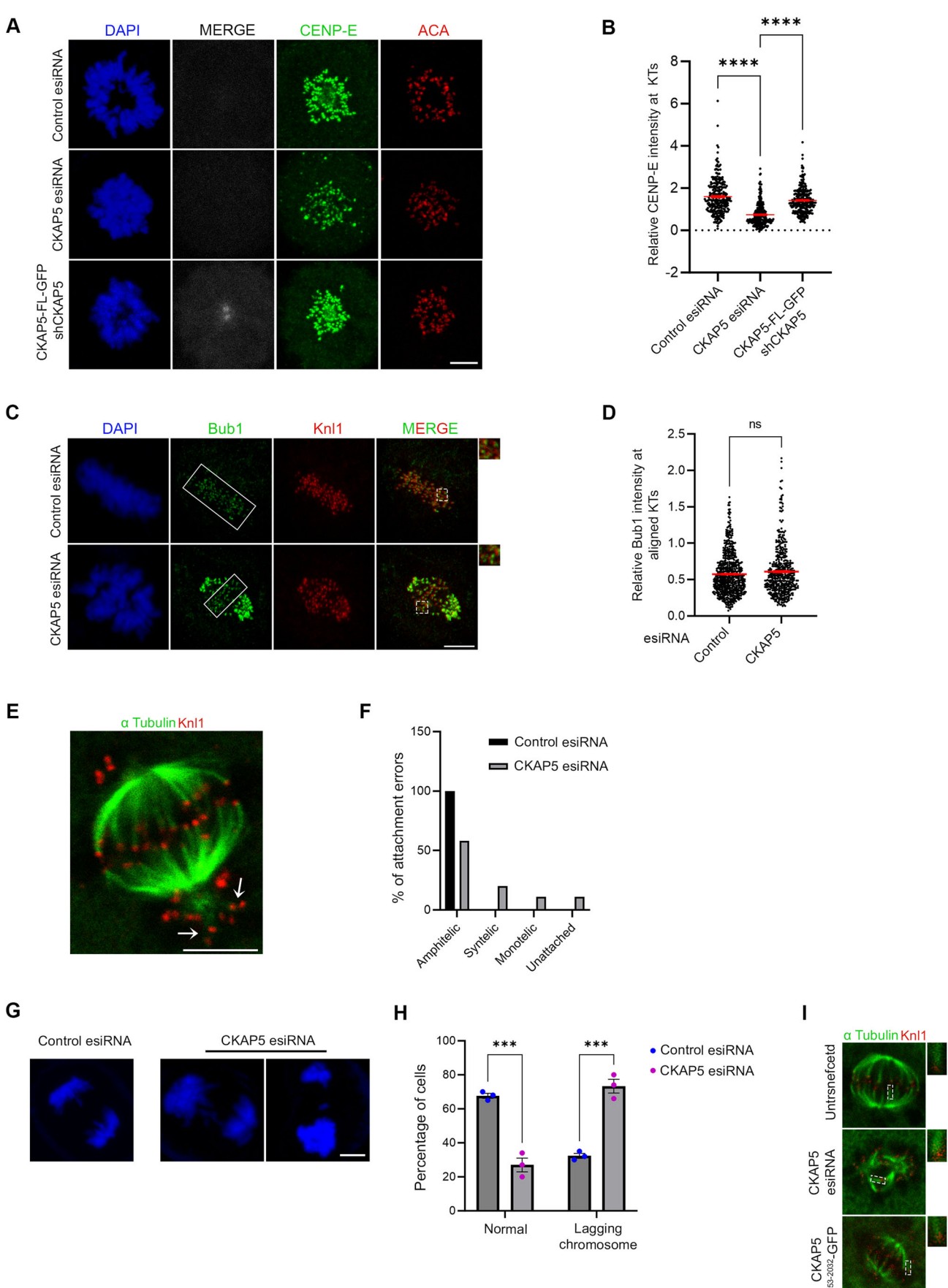

◀ **Figure EV2. CKAP5 depletion results in kinetochore-microtubule attachment errors and lagging chromosomes.**

(**A**) Representative immunofluorescence images showing rescue of CENP-E (green) localization at prometaphase KTs of HeLa Kyoto cells expressing CKAP5-FL-GFP (grey) when compared to CKAP5-depleted cells. ACA (red) was used as KT marker. (**B**) Dot plot showing mean intensity of endogenous CENP-E at these KTs normalized with that of ACA where each dot represents individual KT ($n$ ~270) quantified from multiple cells from three independent experiments. ****$P$ < 0.0001 by one-way ANOVA. (**C**) Representative immunofluorescence image showing no significant change in localization of Bub1 at aligned KTs of control vs. CKAP5-depleted HeLa Kyoto cells. Insets show enlarged view of region marked by dotted boxes in the merge images. (**D**) Dot plots showing mean intensities of Bub1/Knl1 ($n$ ~400 KTs) quantified from individual kinetochores of multiple cells from three independent experiments. (**E**) Representative immunofluorescence image showing syntelic attachments (marked with arrowheads) of polar chromosomes. (**F**) Plot showing percentage of amphitelic, syntelic, monotelic and unattached KTs (144 KT pairs from multiple cells) of pole-proximal chromosomes in CKAP5-depleted HeLa Kyoto cells. (**G**) Anaphase lagging chromosomes shown by CKAP5-depleted HeLa Kyoto cells. (**H**) Graph showing increased percentage of cells showing anaphase laggards upon depletion of CKAP5. More than 100 cells were analyzed from three independent experiments. ***$P$ = 0.0007 by two-way ANOVA. (**I**) Representative immunofluorescence images (single plane) showing cold stable microtubule staining (α Tubulin in green; KNL1 in red) in HeLa Kyoto cells expressing CKAP5 esiRNA and CKAP5$^{853\text{-}2032}$-GFP compared to un-transfected HeLa Kyoto cells. Region in dashed boxes is enlarged in inset. Scale bar of all images = 5 μm. All data represents mean +/- SEM. Data Information: Expression of CKAP5-FL-GFP (grey) rescued CENP-E levels (green) at prometaphase KTs under endogenous CKAP5-depleted condition (**A**). Rectangular boxes show region (aligned KTs) taken for Bub1 intensity analysis. Dotted boxes in merged images showing few KTs enlarged in inset. Bub1 (green) levels showed no difference (**C**). Arrows show KTs of polar chromosomes with syntelic attachments (**E**). DAPI staining of anaphase chromosomes of control and CKAP5-depleted cells. CKAP5-depleted cells showed multiple lagging chromosomes (**G**). Dotted rectangular boxes are enlarged in insets showing increased MT (green) thickness in CKAP5-depleted condition and rescued MT intensity upon expression of CKAP5$^{853\text{-}2032}$-GFP (**I**).

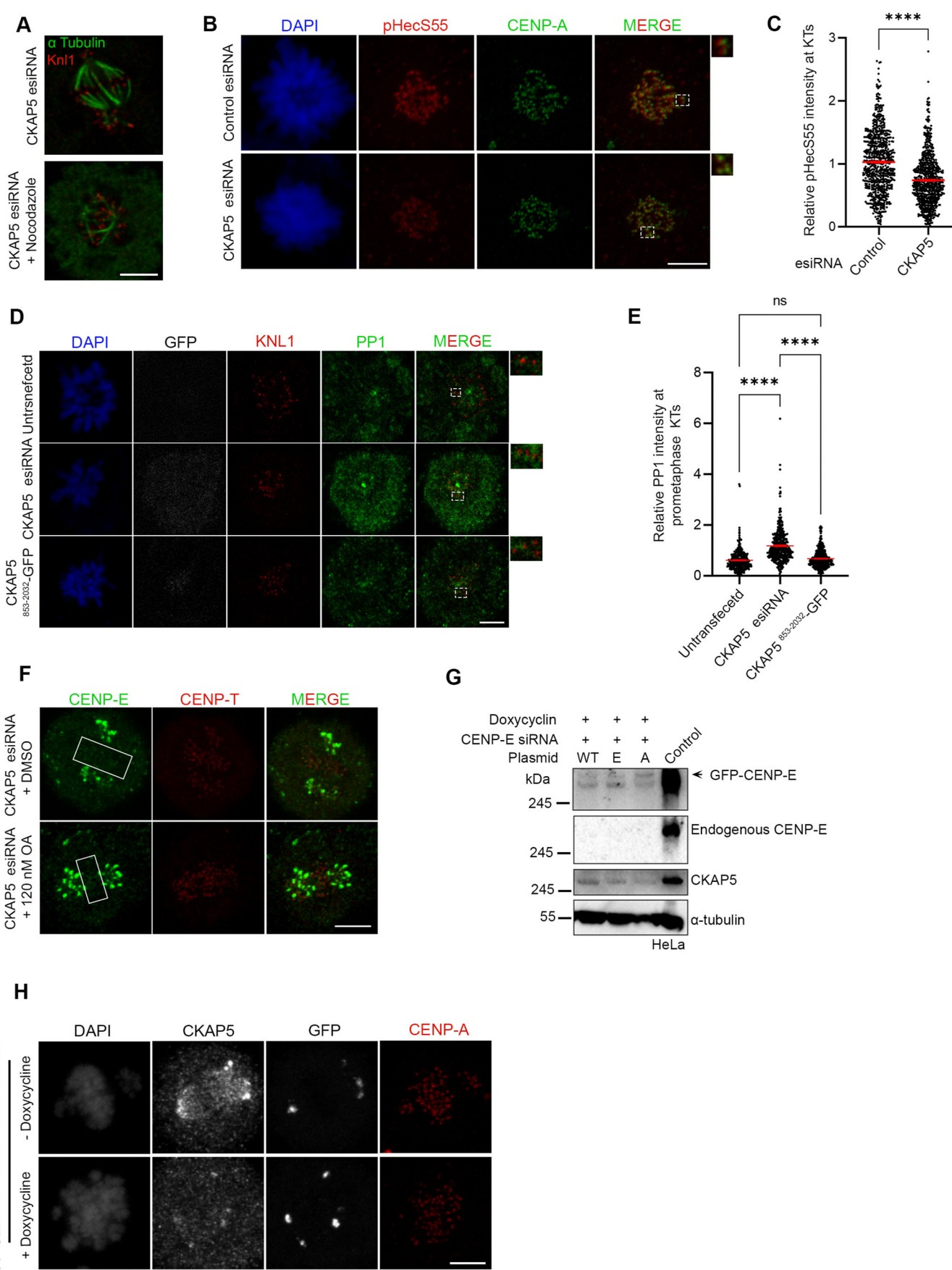

**Figure EV3. CKAP5 depletion leads to increased PP1 activity at the partially aligned kinetochores.**

(A) Representative immunofluorescence image showing partial destabilization of MTs in CKAP5-depleted HeLa Kyoto cells using 3.3 µM nocodazole for 10 min. (α Tubulin in green; KNL1 in red). (B) Immunofluorescence images showing pHecS55 (red) and CENP-A (green) at prometaphase kinetochores of control and CKAP5-depleted HeLa Kyoto cells. Insets show the enlarged view of the dashed boxes. (C) Dot plot showing mean fluorescence intensity of pHecS55 normalized to that of CENP-A quantified from individual kinetochores ($n$ ~700 KTs) of multiple cells from three independent experiments. ****$P < 0.0001$ by Student's $t$ test. (D) Immunofluorescence images (single plane) showing PP1 (green) and KNL1 (red) at prometaphase KTs of HeLa Kyoto cells expressing CKAP5 esiRNA and CKAP5$^{853-2032}$-GFP compared to un-transfected cells. Regions in dashed boxes are enlarged in insets. (E) Dot plot showing mean fluorescence intensity of PP1 from individual kinetochores ($n$ ~400 KTs) normalized with that of KNL1. KTs from individual z planes of multiple cells were analyzed from three independent experiments. ****$P < 0.0001$ by one-way ANOVA. (F) Representative immunofluorescence image showing localization of CENP-E in CKAP5 esiRNA-treated cells incubated with DMSO or 120 nM okadaic acid. Boxes show region of aligned KTs. (G) Western blot showing endogenous CENP-E depletion and doxycycline-induced depletion of CKAP5 in doxycycline-inducible CKAP5 KO HeLa. (WT-Wildtype; A-T422A; E- T422E). (H) Representative immunofluorescence images showing KT localization of GFP-CENP-E-T422A in the presence or absence of doxycycline in doxycycline-inducible CKAP5 KO HeLa cells under endogenous CENP-E depleted background. All data represent mean $+/-$ SEM. Scale bar of all images $= 5$ µm. Data Information: MTs (green) show partial destabilization upon treatment with 3.3 µM nocodazole for 10 min (A). Dotted boxes in merged image are enlarged in insets showing increased levels of pHecS55 (red) at prometaphase KTs upon CKAP5 depletion (B). Dotted boxes in merged image show few KTs that are enlarged in insets. PP1 levels (green) at prometaphase KTs were rescued upon expression of CKAP5$^{853-2032}$-GFP (D). Rectangular boxes show level of CENP-E (green) at aligned KTs. No change in CENP-E levels were observed upon treatment with low concentration of okadaic acid to inhibit PP2A (F). GFP-CENP-E-T422A showed no localization to aligned KTs in the presence or absence of CKAP5 (H).

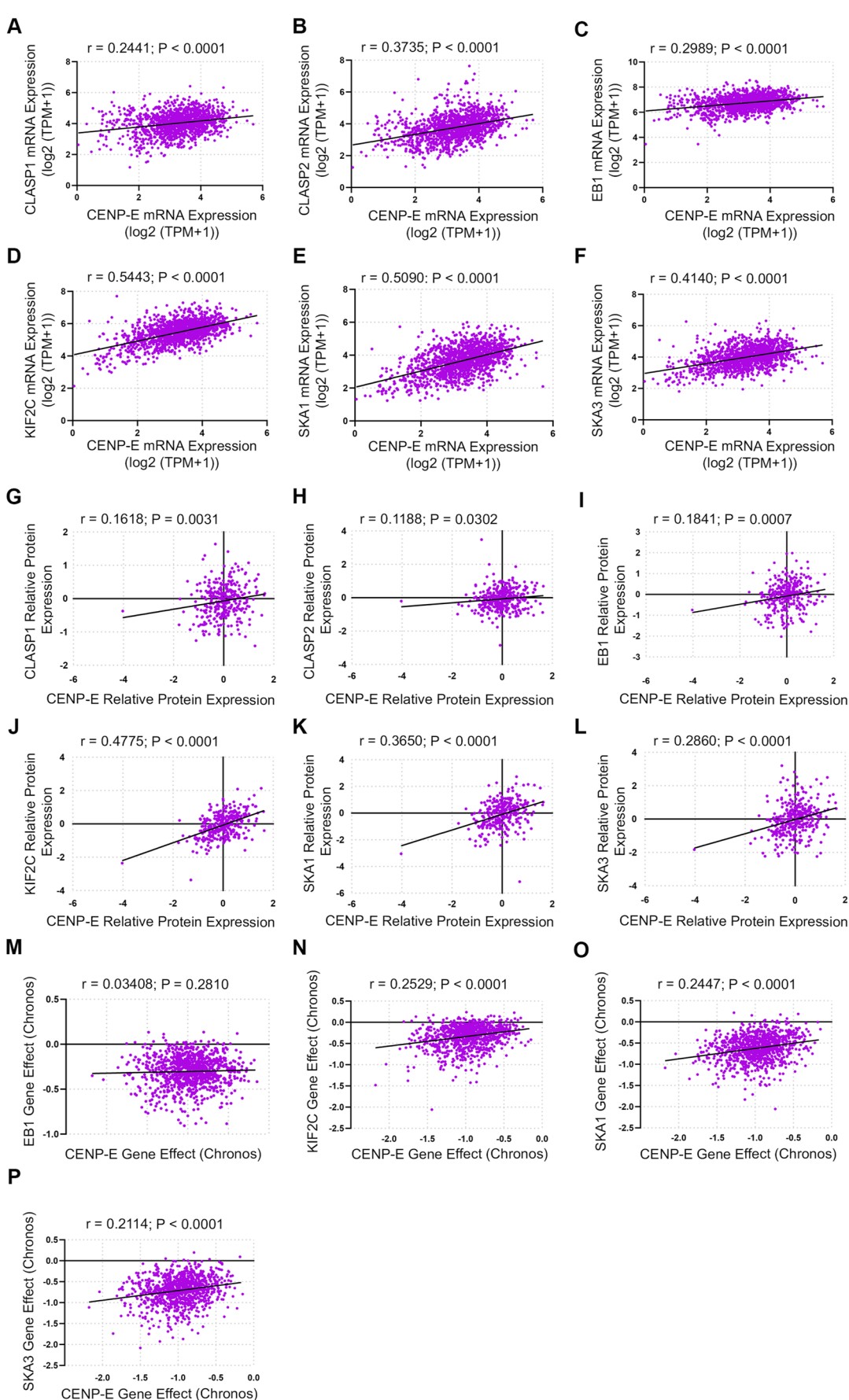

◀ **Figure EV4. Correlation between CKAP5 and CENP-E with respect to other spindle/kinetochore proteins.**

(A–F) Correlation of mRNA expression of CENP-E and multiple spindle/kinetochore proteins. (A) CLASP1, (B) CLASP2, (C) EB1, (D) KIF2C, (E) SKA1, (F) SKA3. The correlation is similar across proteins and similar to the correlation between CKAP5 and CENP-E mRNA expression (Fig. 6E). (G–L) Correlation of protein expression of CENP-E and multiple spindle/kinetochore proteins—(G) CLASP1, (H) CLASP2, (I) EB1, (J) KIF2C, (K) SKA1, (L) SKA3. The correlation is similar across proteins and similar to the correlation between CKAP5 and CENP-E protein expression (Fig. 6F). (M–P) Correlation of sensitivity to CRISPR gene perturbation of CENP-E and multiple spindle/kinetochore proteins. All data is normalized to the mRNA expression of the proliferation marker MKI67 as a covariate using a linear model. (M) EB1, (N) KIF2C, (O) SKA1, (P) SKA3. The correlation is similar across proteins and to the correlation between CKAP5 and CENP-E CRISPR perturbation sensitivity (Fig. 6G). All data is normalized to the mRNA expression of the proliferation marker MKI67 as a covariate using a linear model. Two-sided Pearson's correlation was used for all the data analysis.

