## [Peer Review File · EMBO Reports]

CKAP5 stabilizes CENP-E at kinetochores by regulating microtubule-chromosome attachments

R. Bhagya Lakshmi, Pinaki Nayak, Linoy Raz, Apurba Sarkar, Akshay Saroha, Pratibha Kumari, Vishnu Nair, Delvin Pauly, S. Sajana, Sanusha M. G., Sarit Agasti, Raja Paul, Uri Ben-David, and Tapas Manna

Corresponding author(s): Tapas Manna (tmanna@iisertvm.ac.in)

Review Timeline:

Submission Date:	25th Dec 23
Editorial Decision:	25th Jan 24
Revision Received:	2nd Feb 24
Accepted:	12th Feb 24

Editor: Deniz Senyilmaz Tiebe

Transaction Report: A revised version of this manuscript was transferred to EMBO reports following peer review at another journal.

Responses to the Reviewers:

Reviewer 1:

In the manuscript entitled "CKAP5 regulates microtubule-chromosome attachment fidelity by stabilizing CENP-E at kinetochores", Lakshmi and colleagues show that CKAP5 depletion facilitates PP1 and impairs CENP-E localization at microtubule-attached kinetochores, as well as leads to increased stability of kinetochore-microtubule attachments that results in erroneous chromosome attachments. The results are in line with the proposed role of CKAP5 in correction of erroneous kinetochore-microtubule attachments (Herman et al, Elife 2020). Although the study provides new data on the role of CKAP5 in mitotic error-correction and the regulation of kinetochore-microtubule dynamics, it remains unclear which are the primary effects of CKAP5 depletion and which are the consequent indirect events. The study is for the most part well-conducted, however some important controls are missing and some interpretations of the results should be taken with more caution. Prior to publication the authors should address the following issues.

Q1. Page 7; Figure 1 I, J - Pseudometaphase cells are arrested and their chromosomes spend longer time in mitosis compared to control metaphase cells, which can affect protein levels at aligned kinetochores. Since live-cell imaging has not been performed and immunofluorescence-based analysis cannot distinguish for how long cells have been arrested, it would be more accurate to perform CKAP rescue experiments in mono-asters of EG-5 inhibited cells, as has been done in Figure 1 D-F.

Ans: We thank the reviewer for the suggestion. In the rescue experiment with the different CKAP5 constructs, control cells and all the transfected cells were arrested in G1/S, 12 hours post transfection for 24 hours using thymidine. These cells were released from thymidine block and fixed for immunostaining at 9 hours after release. This ensures that the cells were not arrested at the mitotic stage for a longer time.

Following the suggestion of the reviewer, a rescue experiment was performed in the presence of EG-5 inhibitor, DMA in cells transfected with plasmid containing CKAP5 shRNA and exogenous CKAP5-GFP. The exogenously expressed CKAP5-GFP efficiently rescued CENP-E localization at the KTs in the prometaphase cells. The data (images and quantification plot) are now shown in **Fig. EV2 A and B** (Page 7 and 8).

Q2. Pages 7 and 11; Figures 1 G and 4 E, F - The authors show that CKAP5 depletion-induced CENP-E reduction at kinetochores depends on microtubules and PP1 activity, but it is unclear whether CENP-E is stripped by dynein or simply via the phosphatase pathway. What happens to CENP-E when CKAP5 and dynein are co-depleted/inhibited? What is the effect of PP1 inhibition in the absence of microtubules (nocodazole treatment)?

Ans: We thank the reviewer for this important concern. As per the suggestions, CKAP5 and the dynein adaptor Spindly were co-depleted in HeLa Kyoto cells and it was observed that CENP-E was significantly rescued in the co-depleted condition, suggesting that CENP-E is stripped off from the KTs by dynein in the absence of CKAP5. The new data (images and quantification plot) are now shown in **Fig. EV1 H and I** (Page 7). A recent study showed that phospho-deficient mutant of CENP-E (T422A) does not localize to aligned KTs and its localization was rescued upon depletion of Spindly, suggesting that CENP-E is likely to be stripped off in the dephosphorylated form (Eibes S. et al., 2023, Nature communications, 14:5317). This is

consistent with our finding that PP1 localization at KTs is increased (**Fig 4 A-D**) in CKAP5 depleted condition and that could cause KT removal of CENP-E as we indeed observed (**Fig. 1 A**). We have shown that CENP-E localization at the unattached kinetochores is independent of CKAP5 since CENP-E levels are unaltered in CKAP5 depleted cells under nocodazole treatment (**Fig. 1 G**). Earlier report suggests that PP1 localization to KTs is impaired upon microtubule depolymerization using nocodazole (DeLuca J. et al., 2011, J. Cell Sci. 124 (4): 622-634). Hence, we expect minimal or no effect of PP1 inhibition on CENP-E localization at unattached KTs. Since CKAP5 does not regulate CENP-E localization on un-attached KTs, any effect of PP1 inhibition on CENP-E localization under such conditions specific to un-attached KTs is expected to be CKAP5-independent.

Q3. Page 8; Figure 2 D - The authors claim that they confirmed the direct interaction between CKAP5 and CENP-E using purified GST-CKAP5 and mitotic cell lysates. This is not correct. The direct interaction could be claimed only if purified CENP-E had been used instead of the mitotic lysates.

Ans: We thank the reviewer for this critical comment. Since purification of full-length CENP-E is difficult, we purified the KT targeting domain of CENP-E, MBP-CENP-E²⁰⁵⁵⁻²⁶⁰⁸ and used it for in-vitro pull down to check its interaction with purified GST-CKAP5⁸⁵³⁻²⁰³². We used GST protein as a control. It was observed that GST-CKAP5⁸⁵³⁻²⁰³² interacted with MBP-CENP-E²⁰⁵⁵⁻²⁶⁰⁸ whereas GST protein did not. This suggests direct interaction of CKAP5 and CENP-E. Since the interaction of full-length CENP-E could not be shown and the Reviewer 3 has also raised questions about the role of the interaction in the mechanism of CKAP-5-mediated CENP-E stabilization and the same Reviewer was more in favour of the role of CKAP-5 mediated control of PP1 to regulate CENP-E stabilization, we decided to remove the interaction data.

Figure for referees not shown.

Q4. Page 10, 14, 16, 18; Figure EV 2E - The authors claim that CKAP5 depletion induces syntelic attachments, which is presented only in a supplemental figure graph. Convincing representative microscopy images should accompany this claim.

Ans: As per the reviewer suggestion, a representative image highlighting the attachment error is now added along with the plot of quantification in **Fig. EV2, E and F**.

Q5. Page 11; Figure 4 I, J - The authors show that phospho-mimetic CENP-E T422E has not been completely removed from the aligned kinetochores, suggesting its stabilization at kinetochores. Several important controls are lacking in this experiment. First, this should be

done in the absence of endogenous CENP-E to avoid potential dimerization with the mutant. Second, the behavior of phospho-mimetic CENP-E T422E should be compared to phosphonull CENP-E T422A and wildtype CENP-E.

Ans: We thank the reviewer for the suggestion. We have now performed the experiment using doxycycline inducible CKAP5 KO HeLa cells (generously provided by Ian Cheeseman, Whitehead Institute, MIT). The cells were transfected with GFP-CENP-E constructs (CENP-E WT, T422A, T422E) followed CENP-E 3'UTR siRNA and the results showed that GFP-CENP-E was removed nearly completely in the WT cells from aligned KTs. However, the T422A mutant could not localize to KTs either in the control or CKAP5 KO condition, suggesting that it is inherently defective in localizing to aligned KTs under normal condition. This is consistent with a recent report (Eibes et al., 2023, Nature communications, 14:5317). The data (images and quantification plot) are now shown in **Fig. 4, I – K; Fig. EV3, G and H** (Page 12).

6. Pages 15, 16, 17 (discussion section) - The authors discuss that it is possible that CKAP5 regulates the stability of kinetochore-microtubule attachments and the associated (i.e. indirect) PPI recruitment to kinetochores, which could then (also indirectly) regulate CENP-E localization. However, they conclude:

"Together, our data suggests that CKAP5 is one of the key components that regulate CENP-E localization at aligned KTs after dynein mediated stripping and is also critical for maintaining dynamic kinetochore-microtubule attachment that ensures fidelity of chromosome segregation."

Based on the presented data, a more accurate conclusion would be that CKAP5 regulates kinetochore-microtubule attachment, thereby indirectly regulating PPI-dependent localization of CENP-E. This possibility should be discussed in a more clear way and should also be taken into consideration in the context of the manuscript title, which currently reads: "CKAP5 regulates microtubule-chromosome attachment fidelity by stabilizing CENP-E at kinetochores". Something like "CKAP5 stabilizes CENP-E at kinetochores by regulating microtubule-chromosome attachment" would probably be more accurate.

Ans: Following the suggestions, we have revised the relevant texts in the discussion by describing the possibilities in greater detail (Page 17 and 18). We have also revised the title.

Minor issues:

1. The authors should choose to use either abbreviated or non-abbreviated versions (e.g. kinetochore/KT and microtubule/MT) throughout the manuscript.

Ans: We have revised the relevant texts as per the suggestions.

2. Page 3 - The authors state:

"Removal of CENP-E from the metaphase congressed state has been shown to delocalize chromosomes from the plate leading to chromosome segregation errors (Gudimchuk et al., 2013; Putkey et al., 2002; Weaver et al., 2003)."

Out of these three references, only Gudimchuk et al., 2013 showed the inhibition (not removal) of CENP-E from the metaphase congressed state.

Ans: We thank the Reviewer for alerting this. We have removed the irrelevant references in the revised version.

3. Page 3 - The original work from Ted Salmon's lab should be cited in the context of dynein-dependent removal of CENP-E and corona proteins (Howell et al., JCB 2001).

Ans: We thank the Reviewer for the suggestion. We have added the suggested reference in the revised manuscript.

4. Pages 5, 10, 15 - The authors write that CKAP5 depletion induces stabilization of kinetochore-attached astral microtubule. Microtubules can be either kinetochore-attached or astral, but not both. A proper term would be something like "stabilization of kinetochore-microtubules of polar chromosomes".

Ans: We thank the Reviewer for the suggestion. We have revised the relevant texts as per the suggestions (Page 5 and 10).

5. Page 11; Figure 4E - CENP-E signal seems to be increased both on aligned (attached) and non-aligned (unattached) kinetochores in cell treated with Okadaic acid. Is this true or different brightness/contrast settings have been used?

Ans: There were no different brightness/contrast settings used for the images of aligned vs unaligned KT's in the imaging systems that are used. It is possible that PP1 inhibition leads to increased level of CENP-E phosphorylation and thereby its localization to the KT's is increased.

6. Page 11 - The authors write:

"PP1-mediated dephosphorylation at the Aurora B targeting site Thr 422 of CENP-E facilitates metaphase chromosome alignment (Kim et al., 2010)."

This is not correct. The cited study shows opposite - Aurora kinases-mediated phosphorylation of Thr 422 of CENP-E facilitates metaphase chromosome alignment

Ans: We thank the Reviewer for alerting this. We have revised the relevant texts by describing the mechanism in detail (Page 11).

Reviewer 2:

The correct kinetochore-microtubule attachment is critical for accurate chromosome segregation during mitosis. Multiple mechanisms and regulators collaborate to ensure stable microtubule attachment at kinetochores. The kinesin motor CENP-E has a well-established role in this process. In this current study, Lakshmi et al. discovered a function of the TOG-domain protein CKAP5 in stabilizing CENP-E at kinetochores. They showed that CKAP5 depletion caused mitotic defects in cultured human cells and reduced CENP-E localization at aligned kinetochores. They further dissected the microtubule-dependent interactions between CKAP5 and CENP-E and presented evidence to indicate that their interactions contributed to the stabilization of CENP-E at kinetochores. They also explored the involvement of the phosphatase PP1 in dephosphorylating and delocalizing CENP-E from kinetochores. They then performed computational modeling to demonstrate the feasibility of the proposed mechanisms. Finally, they presented bioinformatics analyses of existing databases to suggest that CKAP5 might be selectively needed in aneuploid cells.

This study contains some interesting new findings. For example, the dependence of CENP-E on CKAP5 at aligned kinetochores is new. The demonstration of a physical interaction between

CKAP5 and CENP-E is also new. On the other hand, the roles of CENP-E and CKAP5 in chromosome alignment are already known. The general importance of phospho-regulation by PP1 at kinetochores is also established. The computational modeling does not offer non-intuitive testable hypothesis. The proposed selective requirement of CKAP5 for aneuploid cell viability lacks experimental verification. Thus, the new findings are rather fragmented and do not rise to the level of a conceptual advance needed for publication in a major journal.

Ans: We thank the reviewer for the critical analysis of our manuscript. In this regard, we would like to highlight the novel aspects of the study. As the reviewer correctly pointed out, our main finding is that CENP-E stabilization at the metaphase-aligned kinetochores is mediated by CKAP5 and we further demonstrate that this is essential for maintaining correct attachment between kinetochores and the microtubules. Although the importance of phospho-regulation by PP1 at KT was known from earlier studies as the reviewer has pointed out, results of our study have unravelled that CKAP5 regulates PP1 recruitment to the kinetochores to ensure appropriate KT-MT attachment. We have also discussed the possible mechanisms of this finding in the relevant sections of “discussion” (Page 16, 17). Another unique observation of our finding is an interesting correlation between aneuploidy and sensitivity to CKAP5 depletion in cancer cells. Our results along with the additional experimental data now included in the revised manuscript also support a critical role of CKAP5 in the prevention of MT-KT attachment errors. We have demonstrated in the revised manuscript that there is increased prevalence of lagging chromosomes in anaphase cells in the CKAP5 depleted condition supporting occurrence of KT-MT attachment errors, specifically, the merotelic attachments (Sacristan et al., 2018, Nature Cell Biology, 20 (7): 800-810). The new data (images and quantification plot) are now shown in **Fig. EV2, G and H** (Page 9).

In addition, our computational model relies on force-dependent turnover of kMT-KT attachments, resulting in distinct spindle phenotypes in control and CKAP5-depleted cells. The role of the outer KT protein CENP-E, is modelled through a spring-like attachment between the KT and kMT tips in our simulation (Thomas et. al., 2016, Nature Communications, 7, 11665). We presume that kMT-KT attachments are prone to detach under high load force. The depolymerizing kMTs from opposite spindle poles apply tension to the KTs through the CKAP5-stabilized CENP-E, destabilizing the erroneous kMT-KT attachments and leading to the formation of a metaphase plate with bioriented chromosomes.

The depletion of CKAP5, and consequently CENP-E, is believed to weaken the force interactions between kMTs and KTs. Impaired kMT-KT interactions fail to generate sufficient load-force on the MT-KT spring like attachments and prematurely stabilize them with error-prone spindle phenotypes. Under such conditions, our simulation successfully recovers the observed error-prone chromosomal phenotype in CKAP5-depleted cells.

Unfortunately, the measurement of forces between KTs and kMTs mediated by CKAP5-stabilized CENP-E at KTs is beyond the scope of our present experiments. Therefore, our consideration of weakened kMT-KT force interactions in CKAP5-depleted cells, leading to the experimentally observed spindle phenotypes with error-prone kMT-KT attachments, introduces a non-intuitive and testable hypothesis. This hypothesis prompts further experiments aimed at directly measuring the tension applied to the kMT-KT interface through CKAP5-CENP-E proteins, offering valuable insights into the mechanical aspects of kinetochore-microtubule interactions. We have now updated the discussion section of the manuscript with relevant texts.

Reviewer 3:

The study by Lakshmi et al reports on an interaction between the TOG domain protein CKAP5 and the mitotic kinesin CENP-E that regulates CENP-E levels at mitotic kinetochores that are attached to microtubules. This is important for preventing merotelic attachments and chromosome congression errors, as also supported by computational modeling. Mechanistically, the authors show that CKAP5 suppresses the amount of PP1 at kinetochores, which enhances the interaction of CENP-E with its kinetochore receptor BubR1.

This is an interesting study with overall good quality data (with a few exceptions) that provides new insights into the regulation of chromosome segregation. I support publication of in the EMBO journal if the authors can address the following issues:

Q1. The model proposed by the authors is not completely in line with the data. If I interpret the data and text correctly, the authors propose that CKAP5 suppresses microtubule stability which in turn suppresses PP1 at kinetochores which in turn enhances the CENP-E-BubR1 interaction and thereby CENP-E stability at attached kinetochores (see also their words in main text, end of paragraph about Fig 4). Yet there are data in the paper that have no apparent fit in this model, while other data to support the model are missing. For example: In figure 2, the authors outline, convincingly, a CKAP5-CENP-E interaction. How does that fit into the model? And where are the data that show that microtubule stability is the mechanism by which CKAP5 impacts kinetochore CENP-E instead of, for example, the direct interaction shown in F2? Can the authors mimic the effect of CKAP5 depletion by over-stabilizing microtubules? Can they rescue CKAP5 depletion by additional suppression of microtubule stability?

Ans: We thank the reviewer for these important points. As suggested, we tried to mimic the MT hyperstability phenotype by treating control cells with 10 μ M paclitaxel and observed that the CENP-E levels at the KTs were not affected. These data (image and quantification plot) are shown in **Fig. 3, H and I**. It is possibly due to impairment of dynein mediated stripping that has been shown previously to be affected substantially by paclitaxel treatment (Forer et al., 2018, Front Cell Dev Biol., 24:6:77; Gurden et al., 2018, Oncotarget, 9(28), 19525-19542). For suppression of MT stability, we treated cells depleted of CKAP5 with nocodazole (3.3 μ M) for 10 minutes and checked for the levels of CENP-E at aligned KTs. There was a significant rescue of CENP-E under this condition, suggesting a role of MT stability on CENP-E localization. However, the rescue was not to the levels of control condition, possibly because presence of CKAP5 is also required for complete stabilization of CENP-E and the CKAP5-CENP-E interaction could play an additional role in that process. The data (image and quantification plot) are shown in **Fig. 3, J and K** (Page 10). In further support of this possibility, we showed that expression of CKAP5 KK/AA mutant that induces MT hyper stability (Herman et al., 2020, ELife, 9, 1–28), led to only about 50% reduction in CENP-E levels at the aligned KTs- please see **Fig. 3 L and M**. However, the mutant could associate with CENP-E as efficiently as the wild type protein. Therefore, it suggests that MT hyperstability is not solely responsible for the effect and the loss of CKAP5-CENP-E interaction could also contribute to the effect (Fig 3 J and K in the earlier version of the manuscript). Since we could not generate stronger data to strengthen the hypothesis of CKAP5-CENP-E interaction mediated CENP-E stabilization at aligned KTs and also, we are agreeing to the comments of the reviewer, we have now removed these data from the revised manuscript.

Q2. Another key part of the model that is not yet sufficiently supported by the data is the PP1 contribution. First, the evidence that CKAP5 depletion increases PP1 at kinetochores requires further support: the images in 4A are not convincing (of note: observing PP1 at kinetochores has been a major challenge for labs, many have resorted to over-expression of exogenous PP1 but even that has its problems). When okadaic acid is added to CKAP5 depleted cells in 4E, not only do the aligned chromosomes have elevated CENP-E but also the unaligned ones have very high CENP-E levels. Hence, it may reflect a general increase in kinetochore protein levels, not one specific to CKAP5 (which predominantly affects the attached/aligned chromosomes). Finally, I don't understand the T422E experiment. Shouldn't T422E mimic a low PP1 state (it was described by the Cleveland lab as a mutant that does not bind PP1)? Yet the authors observe the opposite: low CENP-E at kinetochores. This contradicts the model they propose, but I may be missing something. The experiment, by the way, misses some controls, such as T422A (predicted to have opposite phenotype) and the WT version.

Ans: We have now provided a better resolution representative image showing PP1 localization at aligned KT's in the CKAP5 depleted cells and the PP1 intensity analysis data shown in **Fig. 4 B** are based on analysis of multiple cells (n~1000 KT's) from three independent experiments. Although it is challenging to observe PP1 at KT's using antibodies, we could get reasonably high-quality images by pre-extraction followed by fixation with PFA. In order to rule out the possibility of a general effect of PP1 inhibition on CENP-E localization at KT's in the CKAP5 depleted cells, we checked the effect of PP1 inhibition on CENP-E levels at KT's in control cells and found no difference in its levels as compared to DMSO control. Therefore, it is not a general effect on CENP-E localization upon Okadaic acid (OA) treatment and it is likely to be specific to CKAP5.

Figure for referees not shown.

With regard to the comment on CENP-E T422E localization at KT's, as the Reviewer rightly said, CENP-E T422E mutant localization should mimic low PP1 state. In fact, our previous data showed stabilization of CENP-E T422E. To substantiate the finding further and also following the suggestion of Reviewer # 1, we have checked the KT levels of the T422E mutant under endogenous CENP-E depleted condition in doxycycline-inducible CKAP5 knockout HeLa cells (received as gift from Dr. Iain Cheeseman, Whitehead, MIT) and observed that CENP-E T422E is significantly stabilized at the aligned KT's, similar to what was observed earlier in CKAP5 depleted HeLa Kyoto cells without depletion of endogenous CENP-E. Following the Reviewer's comment, we also performed the missing control experiments for WT and T422A mutant and observed that unlike the T422E mutant, the CENP-E WT failed to

localize to KTs in the CKAP5 knockout cells. The CENP-E T422A mutant, however, could not localize to KTs even in control cells. Therefore, verifying its KT localization under CKAP5 knockout was not possible. The new data (images and quantification plot) are now shown in **Fig 4, I – K** by replacing the earlier data. Additional data for T422A mutant is shown in **Fig EV3, H** (Page 12).

Q3. Figure 6: how specific to CKAP5-CENP-E are the observed correlations? What if the authors do a similar analysis for other spindle/kinetochore proteins? My hunch is that they will show similar correlations (as often with proteins involved in cell division), which would weaken the argument that a correlation in cancer omics data is indicative of a functional connection, other than that both proteins are involved in mitosis.

We thank the Reviewer for raising this point. First, we repeated the analysis after controlling for proliferation using MKI67 mRNA expression levels. All associations between CKAP5, CENP-E and aneuploidy remained highly significant – please see the revised **Fig. 6 and Fig. EV4**. Next, we repeated the correlation analyses with multiple spindle/kinetochore proteins. The results are presented in **EV4**. In short, the correlation of CKAP5 with CENP-E expression (mRNA and protein) is not an outlier – as predicted by the Reviewer, other spindle/kinetochore genes show similar correlations, in line with the co-expression of all of these genes (Page 15). Importantly, however, the association of aneuploidy with sensitivity to CKAP5 depletion is unique. CRISPR-based knockout of CKAP5 is highly associated with aneuploidy, while knockout of other spindle or kinetochore genes is not associated with aneuploidy at all (**Fig.6 D**). Therefore, the cancer omics data further establishes the specific unique role of CKAP5.

Other issues:

- In figure 3, the evidence for merotelic attachment is not convincing. The images show a green signal between sister kinetochores, but it is quite unclear if this represents a merotelic attachment vs, for example, a bridging fiber. Perhaps the authors can better show whether CKAP5 depletion causes lagging chromosomes in anaphase.

Ans: Following the suggestion of the Reviewer, anaphase was induced in control and CKAP5 depleted cells using Mps1 kinase inhibitor (Sacristan et al., 2018, Nature Cell Biology, 20 (7): 800-810). We found increased percentage of cells with lagging chromosomes in anaphase upon CKAP5 depletion. The results with quantification are now shown in **Fig. EV2, G and H** (Page 9).

- I have no expertise in the computational modeling but I had a bit of a hard time understanding what exactly it contributes to the paper. It does not really give testable prediction or new information that the experiments haven't already given.

We appreciate the reviewer for raising this concern. Another reviewer has also noted a similar point. In our model, we introduced a force-dependent turnover of kMT-KT attachments, assuming that erroneous attachments are more prone to breakage under high kMT-KT load force. The molecular interactions of fibrous corona proteins (including CKAP5, CENP-E and other corona protein members) in the kMT-KT interface are modelled through spring-like attachments between KTs and kMTs. A large body of previous studies have also considered such spring-like attachment between kMT tips and KTs (Thomas et. al., 2016, Nature Communications, 7, 11665). Depolymerizing kMTs apply tension to the KTs through binding

KT proteins, destabilizing improper kMT-KT attachments and forming a metaphase plate with bioriented chromosomes.

We hypothesize that the reduced localization of CENP-E at KTs in CKAP5 depleted cells results in a weak kMT-KT coupling, modelled through reduced force strength of the attachment springs between kMTs and KTs in our simulation. Impaired kMT-KT interactions fail to generate sufficient load-force on the kMT-KT spring like attachments and prematurely stabilize them with error-prone spindle phenotypes. Under these conditions, our computational model successfully reproduces the observed chromosomal organization in CKAP5-depleted cells.

Unfortunately, measuring forces between KTs and kMTs mediated by CKAP5-stabilized CENP-E at KTs is beyond the scope of our experiments. Therefore, our consideration of weakened kMT-KT force interactions in CKAP5-depleted cells, resulting in experimentally observed spindle phenotypes with error-prone kMT-KT attachments, introduces a non-intuitive and testable hypothesis. This prompts further experiments to directly measure the tension applied to the kMT-KT interface through CKAP5-CENP-E proteins, providing valuable insights into the mechanical aspects of kinetochore-microtubule interactions. We have now added relevant texts in the discussion section (Page 19).

Dear Tapas,

Thank you for transferring your revised manuscript, which was previously peer-reviewed at another journal. It has now been seen by one of the original referees (referee #1), who evaluated your response to initial concerns of all of the referees. The referee finds that the study is significantly improved during revision and recommend publication.

However, I need you to address the points below before I can accept the manuscript.

- Please submit the manuscript full text as a single MS Word file (including figure legends, expanded view figure legends, tables and their legends and references). The final character count must be clearly indicated on the title page of the manuscript.
- We note that there are currently 8 keywords. However, we can accommodate maximum 5 keywords. Please remove 3 keywords.
- Please add the 'Disclosure Statement and Competing Interests' title before the following statement: 'The authors declare that they have no conflict of interest.'
- We note the following discrepancies between the manuscript and the manuscript submission system regarding the author names: R. Bhagya Lakshmi vs. R Bhagya Lakhshmi; Sanusha M. G vs. M. G. Sanusha.
- As per our format requirements, in the reference list, citations should be listed in alphabetical order and then chronologically, with the authors' surnames and initials inverted; where there are more than 10 authors on a paper, 10 will be listed, followed by 'et al.'. Please remove the DOI's of the published papers, DOIs should only needed for preprints and datasets that have not been published yet. Please see <https://www.embopress.org/page/journal/14693178/authorguide#referencesformat>.
- Please use the data citation format for the previously published datasets cited in the manuscript. Please see <https://www.embopress.org/page/journal/14693178/authorguide#referencesformat> for examples.
- Please fill out and include an author checklist as listed in our online guidelines (<https://www.embopress.org/page/journal/14693178/authorguide>)
- We note that the entire funding info is missing from the manuscript submission system. The funding info from the Acknowledgements section must be entered in the online submission system and must match.
- The EV figures need to be uploaded separately similar to the main figures (one file per figure).
- We note that Figure EV4 and Appendix Table S1 are currently not called out in the text.
- We note an Appendix PDF file containing EV figures, one Appendix Table and some Appendix Supplementary Methods.
- Please remove the EV figures from the Appendix PDF and upload them separately (as mentioned above), and transfer the EV Figure legends to the main manuscript text.
- Please remove the movie legends from the Appendix PDF and provide each movie legend as a readme.txt file and zip the legends with the corresponding movie file.
- We note that there are Appendix Supplementary Methods and Supplementary References. Since we do not have a word of reference limit, please move these sections to the main manuscript.
- Please submit Appendix Table S1 as an Expanded View Table, as it is the only item left in the Appendix File. Please call out the EV Table in the text accordingly.
- Please submit source data as per the email from our Source Data Coordinator Dr. Hannah Sonntag.
- The manuscript sections should be in the following order: Title page - Abstract & Keywords - Introduction - Results - Discussion - Materials & Methods - Data Availability - Acknowledgments - Disclosure Statement & Competing Interests - References - Figure Legends - Tables with legends - Expanded View Figure Legends.
- Our production/data editors have asked you to clarify several points in the figure legends:
 - Please note that the box plots need to be defined in terms of minima, maxima, centre, bounds of box and whiskers, and percentile in the legends of figures 6a-c.
 - Please note that information related to n is missing in the legends of figures 5g, j; m; 6a-c; EV 2h.
 - Please note that n=2 in figure 2f.
 - Please note that the error bars are not defined in the legends of figures 3b, e; 4h, k; 5j; EV 1b; EV 2h.
- *Figure Legends - Comments*
 - Please note that a separate 'Data Information' section is required in the legends of figures 2b, f; 4a, c, e, i-j; 5g, m; EV 1a, e-h, l, n; EV 2a, c, e, g, i; EV 3a-b, d, f, h.
 - Please note that the legends for figures 1b-e is not provided in the sequential manner (legend for figures 1d, e are provided before legend of figures 1b-c and 1c, respectively). This needs to be rectified.
 - Please note that the legends for figures 4b-c is not provided in the sequential manner (legend for figure 4c is provided before legend of figure 4b). This needs to be rectified.
 - Please define the annotated p values ****/*** in the legend of figure EV 2h; EV 3e; as appropriate.
 - Please indicate the statistical test used for data analysis in the legends of figures 2d, 3i; EV 2h; EV 3e; EV 4a-p.
 - Please note that in figure 4f there is a mismatch between the annotated p values in the figure legend and the annotated p values in the figure file that should be corrected.
 - Please note that the scale bar is missing for figure EV 2e.
 - Please note that the dotted white box is not defined in the legend of figure 2g; 3j; 4i-j; EV 1e-f, h. This needs to be rectified.
 - Please note that the dotted yellow box is not defined in the legend of figure 2i; 3l. This needs to be rectified.
 - Please note that the white arrow is not defined in the legend of figure 3d. This needs to be rectified.

- Papers published in EMBO Reports include a 'synopsis' and 'bullet points' to further enhance discoverability. Both are displayed on the html version of the paper and are freely accessible to all readers. The synopsis includes a short standfirst summarizing the study in 1 or 2 sentences (max 35 words) that summarize the paper and are provided by the authors and streamlined by the handling editor. I would therefore ask you to include your synopsis blurb and 3-5 bullet points listing the key experimental findings.
- In addition, please provide an image for the synopsis. This image should provide a rapid overview of the question addressed in the study but still needs to be kept fairly modest since the image size cannot exceed 550 (width) x 300-600 (height) pixels.

Thank you again for giving us to consider your manuscript for EMBO Reports, I look forward to your minor revision.

Kind regards,

Deniz

--

Deniz Senyilmaz Tiebe, PhD
Editor
EMBO Reports

Referee #1:

In the revised version of the manuscript, all my concerns have been appropriately addressed. I want to congratulate the authors and I support publication of this study in EMBO Reports.

All editorial and formatting issues were resolved by the authors.

Dear Tapas,

Thank you for submitting your revised manuscript. I have now looked at everything and all is fine. Therefore, I am very pleased to accept your manuscript for publication in EMBO Reports.

Congratulations on a nice work!

Kind regards,

Deniz

--

Deniz Senyilmaz Tiebe, PhD
Scientific Editor
EMBO Reports

--
